# Assessment of the influential mechanisms of adolescent risk behaviors and protective and risk factors among high school students in Finnmark, Arctic Norway

Shiho Hansen◉°*

Sámi Norwegian National Advisory Unit for Mental Health and Substance Use, Finnmark Hospital Trust, Karasjok, Norway

too The author contributed equally to this work.
* Shiho.Hansen@finnmarkssykehuset.no

## Abstract

Adolescent risk behaviors, including alcohol use and antisocial behaviors, remain a public health concern in Finnmark, Arctic Norway. However, the mechanisms linking risk and protective factors to these behaviors remain underexplored. This study examines the influence of individual and environmental factors—such as family, peers, school, and local contexts—on adolescent alcohol use and antisocial behaviors. Using data from the 2021 Ungdata survey (N = 2,129 high school students), Partial Least Squares Structural Equation Modeling was applied to identify key associations. The results indicate that unstructured peer interactions, particularly spending evenings with friends, are strongly associated with both alcohol use and antisocial behaviors. In contrast, time spent on online gaming was associated with reduced risk of these behaviors, while social media use was positively linked to alcohol consumption. Experiences of sexual harassment, as well as other co-occurring risk behaviors such as smoking, drug use, bullying, and aggression, were consistently associated with increased engagement in both alcohol use and antisocial behaviors. Family and school environments showed limited associations with alcohol use but were linked to antisocial behaviors. Socioeconomic status and gender were not significantly related to either outcome, while depressive symptoms and religiosity showed selective associations, primarily with alcohol use. The findings underscore the need for peer-oriented preventive strategies and consideration of digital media exposure in addressing adolescent alcohol use and antisocial behavior. Future research should incorporate ethnicity-related variables to enhance contextual understanding of adolescent behaviors in the Finnmark region.

**Data availability statement:** The data for this study are available from the Norwegian Agency for Shared Services in Education and Research (Sikt). Due to ethical and legal restrictions, access to the dataset is limited and requires prior approval. The dataset contains sensitive information and is governed by data protection regulations. Consequently, approval from both Sikt and Norwegian Social Research (NOVA) is required for access and use of the data for research purposes. Researchers interested in accessing the data may submit a request through Sikt's Survey Bank at the following link: https://surveybanken.sikt.no/no/study/NSD3157/1?file=9c72d168-67ae-4c15-82d9-738535771af9/4&type=studyMetadata.

**Funding:** "This research was supported by the internal research fund of Finnmark Hospital Trust. Additionally, the study was made possible through research time approved by my affiliated institution, the Sámi Norwegian National Advisory Unit for Mental Health and Substance Use (SANKS). None of the funding providers had any role in the data analyses and interpretation, nor had they any right to approve or disapprove the writing and publication of the manuscript. There was no additional external funding received for this study."

**Competing interests:** The author has declared that no competing interests exist.

## Background

Adolescents in Finnmark County have shown limited progress in reducing risk behaviors, such as substance use and antisocial behaviors [1,2]. These behaviors can lead to adverse health and socioeconomic outcomes later in life [3,4]. Among risk behaviors, substance use, such as alcohol use, can increase the risk of long-term consequences, including mental health disorders, academic underachievement, substance use disorders, and higher rates of addiction if used regularly during this developmental period [5]. Similar to substance use, antisocial behaviors, such as rule violations and aggressive behaviors, also have adverse consequences and health outcomes in adulthood, including low educational achievement, unemployment, alcohol and drug dependence, criminality, and psychosocial malfunctioning [6].

A trend analysis among high school students in Finnmark showed that the prevalence of substance use declined over time, while antisocial behaviors, such as evading payments in shops or on public transportation, have increased over time [2]. Although the prevalence of alcohol use among adolescents in Norway has decreased in line with international trends, this behavior remains vulnerable to fluctuations based on contextual factors [7]. Given these concerns, integrating a public health perspective is essential when examining adolescents' risk behaviors, such as alcohol use and antisocial behaviors.

### Alcohol use among norwegian adolescents

A study of ethnic Norwegian and ethnic minority adolescents in the capital Oslo region showed that parental education, parent–adolescent relationships, and mental health status were significantly associated with binge drinking, in addition to ethnicity [8]. A study on heavy episodic drinking in three Nordic countries (Finland, Norway, and Sweden) found that a decline in adolescent daily smoking, perceived access to alcohol, and increased parental control were associated with a decline in heavy episodic drinking, whereas truancy, going out with friends, and engaging in sports were not [7]. Another cross-sectional study on anxiety, depressive symptoms, and alcohol use among high school students in Norway showed that depressive symptoms were associated with earlier onset of alcohol use, more frequent consumption, and higher rates of intoxication among adolescents, while anxiety symptoms were only associated with alcohol consumption among female adolescents [9]. Furthermore, a study comparing risk behaviors among young people who are not in education, employment, or training (NEET) with high school students found a higher prevalence of several health-risk behaviors among NEET youth. Being NEET was associated with increased odds of cannabis use, smokeless tobacco use, and smoking cigarettes, but no differences in alcohol intoxication [10].

In addition to studies targeting the country overall or focusing on the capital region, several investigations have explored the factors influencing substance use among adolescents in Northern Norway. The North Norwegian Youth Study (NNYS/Ung I Nord), a longitudinal epidemiological study conducted in the three northernmost counties of Norway (Nordland, Troms, and Finnmark) during 1994–1995 and 1997–1998, examined ethnic identity among Sámi and non-Sámi adolescents and

its influence on risk behaviors [11]. Literature from this study found that non-Sámi adolescents engaged in more drinking behavior, while Sámi adolescents reported greater concerns from friends and family about their drinking. Although Sámi youth exhibited less drinking and higher abstinence, aligning with their parents' drinking habits [12], the highest abstinence rates were found among Sámi parents in the Sami Highland [12,13]. Another study investigating ethnic differences in abstinence, drinking frequency, and context found that religious factors partially explained the association between Sámi ethnicity and drinking behavior. Specifically, religious factors acted as an independent protective factor, linked to lower drinking rates and higher abstinence among Sámi youth and young adults [14]. Additionally, weaker Sámi cultural orientation, reflected by favoring assimilation and residing in assimilated ethnic contexts, was associated with increased substance use, more often in late adolescence than in young adulthood [15].

## Antisocial behavior among norwegian adolescents

Although the number of studies is relatively small compared with the one that examine the influential factors of substance use, several research showed the association between antisocial behaviors and influential factors among adolescents in Norway. Celik [16] examined the determinants of antisocial behavior among high school students in Norway, showing bullying, hate speech, and sexual harassment were significant predictors of antisocial behavior, while positive family and school environments reduced it. Socioeconomic factors, such as urban living and low parental education, were also associated with higher antisocial behavior. A longitudinal study found that low social competence at age 13 predicted antisocial behavior at age 15 [17]. Although social competence and antisocial behavior are negatively correlated, they are distinct dimensions, suggesting targeted prevention efforts are needed [17]. Study that analyzed increases in adolescent physical fighting between 2015 and 2018 in Norway showed a rise in physical fighting among Norwegian adolescents between 2015 and 2018 was linked to unsupervised leisure activities, cannabis use, and digital media consumption [18].

## Finnmark's unique context and adolescent risk behaviors

Studies in Norway showed some associations between risk and protective factors and substance use and antisocial behaviors. However, those study targeted Norway or Northern Norway as a whole and little is targeted solely to Finnmark.

Finnmark, the northernmost county in Norway, has distinct sociocultural and multiethnic contexts compared to those in middle and southern Norway. This includes populations such as the indigenous Sámi (Sámi-speaking peoples inhabiting the Sápmi region) and Kven (descendants of people who migrated to Northern Norway from Northern Finland and Northern Sweden in the 18th and 19th centuries), as well as people from Northern Finland and Northern Sweden, and the intermarriage among these groups [19,20]. This unique multiethnic context adds another layer of complexity to the understanding of adolescent health. Studies conducted in other circumpolar regions, such as Alaska and northern Canada, have highlighted similar patterns, where indigenous youth are disproportionately affected by mental health issues like depression, anxiety, and suicide ideation [21]. Additionally, compared with other regions of Northern Norway, the population in Finnmark is small, dispersed, and isolated [20]. These differences may be related to the distinct patterns of risk and protective factors influencing alcohol use and antisocial behaviors among adolescents in Finnmark, including family, peers, school, and community.

## Theoretical and contextual framework

This study is anchored in Bronfenbrenner's Social-Ecological Model [22], which emphasizes the interplay between individual behaviors and broader environmental influences. This theoretical framework allows for an exploration of how personal, relational, and community-level factors interact to shape adolescent risk behaviors, such as alcohol use and antisocial behavior. The model's dynamic perspective provides a foundation to understand both risk and protective factors influencing adolescent health outcomes in the unique sociocultural and multiethnic context of Finnmark.

At the individual level, the social-ecological model highlights personal factors that significantly impact adolescent behaviors. Socioeconomic status, mental health conditions (such as depression and anxiety), and lifestyle factors (including inactivity) play a crucial role in determining susceptibility to risk behaviors like substance use and antisocial activities [7–10,16,17]. Adolescents facing mental health challenges are more likely to engage in substance use and display antisocial behaviors, underscoring the need for targeted interventions addressing psychological well-being.

The relational level focuses on close social connections, including family relationships, peer influences, and school environments. These elements function as either protective buffer against or contributors to risk behaviors, depending on their structure and quality. Studies indicate that strong parent-adolescent relationships and high parental control are protective against substance use, while negative peer influences and lack of supervision can exacerbate risk behaviors [8,11–15]. Similarly, a positive school climate reduces the likelihood of antisocial behavior, whereas bullying, hate speech, and peer victimization contribute to increased delinquency [16,17].

The broader contextual background of Finnmark plays a critical role in shaping adolescent risk behaviors. As the northernmost county in Norway, Finnmark has a small and geographically dispersed population [19,20]. This demographic structure limits adolescents' access to essential resources, extracurricular opportunities, and support networks, potentially heightening their vulnerability to risk behaviors [23]. Furthermore, Finnmark's multiethnic composition—including indigenous Sámi, Kven, and other northern European populations—adds another layer of complexity to adolescent health outcomes. Research suggests that Sámi youth exhibit lower alcohol consumption rates, influenced by strong parental modeling and religious factors [12–14]. However, weaker Sámi cultural orientation, marked by assimilation tendencies, has been linked to increased substance use in late adolescence [15]. The intersection of ethnicity, cultural identity, and environmental constraints necessitates a nuanced approach to understanding adolescent risk behaviors in Finnmark.

Additionally, comparative trends underscore the region's distinctive challenges [2]. While Norway has generally seen a decline in adolescent substance use, antisocial behaviors such as vandalism and fare evasion have risen in Finnmark [2]. These trends highlight the specific vulnerabilities and contextual factors at play in the region.

### Purpose of the study

Given these contextual and theoretical frameworks, this study aims to investigate the associations between risk and protective factors with risk behaviors among high school students in Finnmark. Specifically, the study aims to investigate how individual characteristics (e.g., socioeconomic status), other individual factors (e.g., depressive thoughts, negative experiences, and digital media use) and environmental factors (e.g., family, school, peers, and local area) are associated with alcohol use and antisocial behaviors.

## Data and methods

### Data

**Data source and samples.** This study used cross-sectional data from the Ungdata survey. Ungdata is a national data collection scheme on health, well-being, and risk behaviors among adolescents in Norway [24]. Norwegian Social Research (NOVA) is responsible for overall coordinating, while the Regional Drug and Alcohol Competence Centers (KoRus) are responsible for conducting the surveys at the municipal level. Since 2014, the survey has been conducted every three years in both junior high and high schools across Norway. The questionnaire consists of a core module, which remains consistent across all survey waves, along with optional questions that municipalities can select based on their specific interests and needs. The questionnaires cover various aspects of adolescents' lives, such as free time use, physical activity, social media use, and environmental factors surrounding adolescents, including family, friends, school, and neighbors. Additional areas included substance use, anti-social behaviors, depressive thoughts, and perceptions of health and future [25].

 

The surveys were administered anonymously online during school hours. For further details on Ungdata, refer to their website [24]. This study used the data collected in 2021 from high school students in Finnmark. The response rate was 73% [1]. Reasons for non-response included student absence on the day of the survey or a refusal to participate. A total of 2,129 high school students in Finnmark joined the survey. Observations from participants who answered only the first few questions were excluded from the analysis, resulting in a final sample of 2,086 high school students.

The distribution of participants by grade level was as follows: 919 (44.1%) in VG1 (1st year of high school), 737 (35.3%) in VG2 (2nd year), 422 (20.2%) in VG3 (3rd year), and 8 (0.4%) with missing grade information. Among them, 1,066 (51.1%) were female, 978 (46.9%) were male, and 42 (2.0%) had missing gender data. The dataset was obtained from the Survey Bank of the Norwegian Agency for Shared Services in Education and Research, or Sikt [26]. While the Ungdata survey did not collect exact age data, the grade levels correspond to typical age ranges in the Norwegian school system: students in VG1 are generally 16 years old, those in VG2 are 17, and those in VG3 are 18. This three-year structure of *videregående skole* (high school education) follows the completion of junior high school and is outlined in § 5−1 of the Norwegian Education Act [27].

**Sample size.**  This study utilized Partial Least Squares Structural Equation Modeling (PLS-SEM) to examine the relationships between variables. A traditional rule-of-thumb method for estimating the sample size in PLS-SEM is the "10-times rule," which suggests that the sample should be at least 10 times the number of estimated parameters. However, this approach may lead to inaccuracies in complex models. More precise methods, such as the inverse square root method and the gamma-exponential method, have been proposed [28]. The inverse square root method is widely used due to its simplicity. Assuming a common power level of 80% and a significance level of 5%, the minimum sample size is calculated as follows: $N > (2.486/(|\beta|_{min}))^2$ In social science studies, it is often acceptable to have low path coefficients (for example $|\beta|_{min}$ is 0.1) that are still statistically significant when using bootstrapping techniques [29]. Assuming a minimum path coefficient $|\beta|_{min}$ 0.1, the required sample size is: and $N > (2.486/0.1))^2 = 618.02$. Thus, the minimum sample size required for analysis will be 619, assuming $|\beta|_{min}$ is more than 0.1.

**Measures.**  This study utilized items from both the core module and optional sections of the Ungdata survey. Only items that were administered across all relevant grade levels were retained for analysis to ensure consistency in the dataset. In total, 59 variables were used for data analysis, with most items measured on Likert-type scales. A total of 24 composite variables were constructed to reflect broader behavioral or contextual domains. These were developed by aggregating conceptually related survey items within thematic categories of the Ungdata questionnaire. For example, "alcohol use" represents risky drinking behaviors and includes three items (e.g., whether the respondent has consumed alcohol or enough to feel clearly intoxicated, measured by items *balko1* to *balko3*). Similarly, "antisocial behavior" comprises five items (e.g., shoplifting, staying out all night without parental knowledge, skipping school, measured by *banti1* to *banti5*).

All composite constructs were modeled using PLS-SEM. This approach calculates construct scores as weighted linear combinations of their indicators, following a three-step procedure: (1) standardization of indicators, (2) estimation of outer loadings (i.e., bivariate correlations between indicators and their composite variable), and (3) computation of final construct scores [30]. Composite variables in PLS-SEM are theory-driven, derived from both conceptual coherence and empirical validation.

Some variables, including "socioeconomic status (SES)" and "religion," were included as single-item indicators and treated as directly measured variables rather than composites. SES was measured using a composite index specifically developed for the Ungdata survey [31,32], combining three components: parental education, number of books in the home, and family affluence. Family affluence was assessed using four items adapted from the Family Affluence Scale [33]: car ownership, having a private bedroom, family vacations in the past year, and the number of computers in the household. These were averaged into a single SES score ranging from 0 to 3. To adjust for age-related variability in adolescents' knowledge of family resources, SES quintiles were calculated separately by grade level.

In order to maintain consistent interpretability across variables, a reverse coding system was applied where necessary. This ensured that higher values reflected more problematic or undesirable conditions. For instance, the item "I often don't want to go to school" [*Jeg gruer meg ofte til å gå på skolen*], originally rated on a four-point Likert scale (1 = "totally disagree" to 4 = "totally agree"), was reverse coded so that lower scores indicated greater school-related distress.

An overview of all composite and measured variables included in the PLS-SEM model is presented in Table 1, which also includes the original item wordings and corresponding response options.

## Statistical methods

**Partial Least Squares Structural Equation Modeling (PLS-SEM).** This study utilized PLS-SEM to examine the relationships between exogenous and endogenous variables. PLS-SEM is a robust multivariate statistical technique commonly applied in disciplines such as the social sciences and health research. Structural Equation Modeling (SEM) encompasses two primary analytical approaches: covariance-based SEM (CB-SEM) and PLS-SEM.

A key advantage of PLS-SEM lies in its flexibility regarding data characteristics and model complexity [30]. Unlike CB-SEM, PLS-SEM is particularly well-suited for studies with small to medium sample sizes. Additionally, it does not require the data to conform to a normal distribution, enabling effective analysis of non-normal, skewed, or kurtotic datasets. PLS-SEM is also adept at estimating complex models comprising multiple constructs, indicators, and interrelationships, without imposing strict statistical assumptions.

Another notable strength of PLS-SEM is its robustness to measurement issues [30]. It demonstrates greater tolerance for measurement errors in observed indicators compared to CB-SEM, thereby enhancing the reliability of results even when data quality is less than optimal. Furthermore, PLS-SEM employs bootstrapping techniques to evaluate the significance of path coefficients within the model. This non-parametric method facilitates rigorous statistical testing without relying on distributional assumptions, providing researchers with greater confidence in their findings.

**Model development.** The conceptual model for this study was developed to explore the interplay between adolescent risk behaviors and a range of individual, social, and contextual factors. The model is grounded in Bronfenbrenner's Ecological Systems Theory [22], which posits that adolescent behavior is shaped by the dynamic interplay between individual characteristics, environmental contexts, and broader societal influences. Variable selection was further guided by empirical studies [6–18,34–37] and the Ungdata research framework [25], which collectively emphasize the multidimensional nature of adolescent risk and protective factors.

Within this framework, alcohol use and antisocial behaviors are classified as risk behaviors, as they encompass actions that may result in negative consequences for adolescents' health, well-being, and social development. Other variables in the model are conceptualized as risk and protective factors, which either increase the likelihood of engaging in risk behaviors or serve as mitigating influences that promote resilience. These factors include psychosocial background that covers social (SES, religion) and psychological (depression, future orientation) components, free-time activities such as physical activity and gaming or social media use, and negative experiences, including being bullied. Additionally, environmental influences, such as parental relationships, peer interactions, linking school, and neighborhood context, are incorporated into the model. Other risk behaviors, such as tobacco and nicotine product use and aggressive behaviors, are also considered to explore potential co-occurring patterns.

Based on these theoretical and empirical underpinnings, the model incorporated both composite variables (as described in the "Measures" section) and directly measured variables. The composite variables were integrated into the PLS-SEM framework using standardized procedures for indicator aggregation and weight estimation. To ensure the reliability and validity of the measurement model, standard evaluation procedures were applied, including assessments of outer loadings, composite reliability, convergent validity (via average variance extracted), and discriminant validity. The structural model was then examined to test hypothesized paths between constructs and to explore indirect and direct effects among predictors and outcomes. Further details are provided in the "Data Analysis" section.

**Table 1. Overview of composite and measured variables used in the PLS-SEM model.**

| Composite variable | Measured variable | Questions | Response options (coding scheme) |
|---|---|---|---|
| **Outcome** | | | |
| Alcohol use | balko1 | Do you ever drink any kinds of alcoholic drinks? | 1 = Never, 2 = I´ve just tried tasting them a few times, 3 = Occasionally, but less than once a month, 4 = Generally 1–3 times a month, 5 = Every week |
| | balko2 | How old were you the first time you drank so much alcohol? | 1 = 18 years or older, 2 = 17 years, 3 = 16 years, 4 = 15 years, 5 = 14 years, 6 = 13 years, 7 = Under 13 years |
| | balko3 | How many times have you had so much to drink that you felt clearly intoxicated over the past year (the past 12 months)? | 1 = Never, 2 = Once, 3 = 2–5 times, 4 = 6–10 times, 5 = 11 or more |
| Anti-social behavior | How many times have you done any of the following things over the past year (the past 12 months)? | | |
| | banti1 | take something from a shop without paying | 1 = Never, 2 = Once, 3 = 2–5 times, 4 = 6–10 times, 5 = 11 or more |
| | banti2 | not paid for the cinema, sporting events, bus and train tickets, etc. when you sholud have paid | 1 = Never, 2 = Once, 3 = 2–5 times, 4 = 6–10 times, 5 = 11 or more |
| | banti3 | spent the whole night away from home without your parents knowing where you were | 1 = Never, 2 = Once, 3 = 2–5 times, 4 = 6–10 times, 5 = 11 or more |
| | banti4 | skipped school | 1 = Never, 2 = Once, 3 = 2–5 times, 4 = 6–10 times, 5 = 11 or more |
| | banti5 | Consciously cheated on a test or assignment that was graded | 1 = Never, 2 = Once, 3 = 2–5 times, 4 = 6–10 times, 5 = 11 or more |
| **Psychosocial Background** | | | |
| Socioeconomic status | Ses | Socioeconomic status | 0-3 |
| Religion | Relig | How much does religion mean for how you live your daily life? | 1 = It is very important, 2 = Religion means quite a lot for how I live my daily life, = Religion means little for how I live my daily life, 4 = Religion has no significance for how I live my life |
| Future outlook | What do you think your future will be like? Do you think that you: | | |
| | fram1 | will have a good, happy life? | 1 = Yes, 2 = Don´t know, 3 = No |
| | fram2 | will complete upper secondary school in the future? | 1 = Yes, 2 = Don´t know, 3 = No |
| Depressionive symptom | During the past week, have you been affected by any of the following issues | | |
| | depr1 | felt that everything is a struggle | 1 = Been affected a great deal, 2 = Been affected quite a lot, 3 = Not been affected much, 4 = Not been affected at all |
| | depr2 | had sleep problems | 1 = Been affected a great deal, 2 = Been affected quite a lot, 3 = Not been affected much, 4 = Not been affected at all |
| | depr3 | felt unhappy, sad or depressed | 1 = Been affected a great deal, 2 = Been affected quite a lot, 3 = Not been affected much, 4 = Not been affected at all |
| | depr4 | felt hopelesness about the future | 1 = Been affected a great deal, 2 = Been affected quite a lot, 3 = Not been affected much, 4 = Not been affected at all |
| | depr5 | felt stiff or tense | 1 = Been affected a great deal, 2 = Been affected quite a lot, 3 = Not been affected much, 4 = Not been affected at all |
| **Adverse Experiences** | | | |
| Covid-19 | covid1 | Has the coronavirus pandemic negatively affected your life? | 1 = No, not at all, 2 = Yes, a little, 3 = Yes, somewhat, 4 = Yes, a lot, 5 = Yes, very much |
| | covid2 | I have felt more unhappy, sad, or depressed than before the coronavirus. | 1 = No, not at all, 2 = Yes, a little, 3 = Yes, somewhat, 4 = Yes, a lot, 5 = Yes, very much |
| | covid3 | There has been more arguing in my family than before the coronavirus. | 1 = No, not at all, 2 = Yes, a little, 3 = Yes, somewhat, 4 = Yes, a lot, 5 = Yes, very much |

*(Continued)*

**Table 1.** (Continued)

| Composite variable | Measured variable | Questions | Response options (coding scheme) |
|---|---|---|---|
| Being bullied | mobb1 | Are you sometimes teased, threatened or frozen out by other young people at school or in your free time? | 1 = Never, 2 = Almost never, 3 = Yes, around once a month, 4 = Yes, around once a fortnight, 5 = Yes, around once a week, 6 = Yes, several times a week |
| | mobb2 | Are you bullied, threatened, or excluded online? | 1 = Never, 2 = Almost never, 3 = Yes, around once a month, 4 = Yes, around once a fortnight, 5 = Yes, around once a week, 6 = Yes, several times a week |
| | mobb3 | Has another young person hit, kicked, shaken you hard, pulled your hair, or similar? | 1 = Never, 2 = 1 time, 3 = 2–5 times, 4 = 6 or more |
| | mobb4 | Has another young person threatened, attacked, or robbed you with objects? | 1 = Never, 2 = 1 time, 3 = 2–5 times, 4 = 6 or more |
| Sexual harassment | sextrak1 | Someone touched you in a sexual way against your will. | 1 = Never, 2 = 1 time, 3 = 2–5 times, 4 = 6 or more |
| | sextrak2 | Someone hurtfully called you a whore, gay, or other words with sexual content. | 1 = Never, 2 = 1 time, 3 = 2–5 times, 4 = 6 or more |
| | sextrak3 | Someone spread negative sexual rumors about you. | 1 = Never, 2 = 1 time, 3 = 2–5 times, 4 = 6 or more |
| | sextrak4 | Someone shared pictures or videos of you where you are naked or participating in sexual acts without your consent. | 1 = Never, 2 = 1 time, 3 = 2–5 times, 4 = 6 or more |
| **Leisure Time Activities** | | | |
| Gaming | Digi | How much time to you spend on playing computer games/video games? | 1 = No time, 2 = Less than 30 mintes, 3 = 30 minutes – 1 hour, 4 = 1–2 hours, 5 = 2–3 hours, 6 = More than 3 hours |
| Social media use | Sms | How much time to you spend on social media (Facebook, Instagram, etc.)? | 1 = No time, 2 = Less than 30 mintes, 3 = 30 minutes – 1 hour, 4 = 1–2 hours, 5 = 2–3 hours, 6 = More than 3 hours |
| Physical activity | Fysak | How often do you participate in other kinds of organised physical activity (dance, martial arts, etc.)? | 1 = At least 5 times a week, 2 = 3–4 times a week, 3 = 1–2 times a week, 4 = 1–2 times a month, 5 = Rarely, 6 = Never |
| **Environmental Factors** | | | |
| Parental monitoring | fam1 | Felt loved by my parents or guardians. | 1 = All the time, 2 = Often, 3 = Some of the time, 4 = Rarely, 5 = Not at all |
| | fam2 | My parents usually know where I am, and who I'm with, in my free time | 1 = Very true, 2 = Quite true, 3 = Not very true, 4 = Not at all true |
| | fam3 | My parents know most of the friends I hang out with in my free time | 1 = Very true, 2 = Quite true, 3 = Not very true, 4 = Not at all true |
| | fam4 | My parents are very interested in my life. | 1 = Very true, 2 = Quite true, 3 = Not very true, 4 = Not at all true |
| Poor relationships with parents | rfam1 | I try to keep most of my free time activities hidden from my parents. | 1 = Not at all true, 2 = Not very true, 3 = Quite true, 4 = Very true, |
| | rfam2 | There is often arguing between the adults in my family. | 1 = Not at all true, 2 = Not very true, 3 = Quite true, 4 = Very true, |
| Friendship support | venn1 | Do you have at least one friend who you trust completely and who you can tell absolutely anything? | 1 = Yes, definitely, 2 = Yes, I think so, 3 = I don´t think so, 4 = There´s nobody I would call a friend at the moment |
| | venn2 | Do you have someone to be with during your free time? | 1 = Yes, definitely, 2 = Yes, I think so, 3 = I don´t think so, 4 = There´s nobody I would call a friend at the moment |
| | venn3 | Do you have someone to be with during recess at school? | 1 = Yes, definitely, 2 = Yes, I think so, 3 = I don´t think so, 4 = There´s nobody I would call a friend at the moment |
| Spending evenings with friends | Bvenn | Think about the last week (last 7 days). How many times have you spent most of the evening out with friends/mates | 1 = Never, 2 = 1 time, 3 = 2–5 times, 4 = 6 or more |

*(Continued)*

| Composite variable | Measured variable | Questions | Response options (coding scheme) |
|---|---|---|---|
| Liking school | skole1 | I enjoy school | 1 = Totally agree, 2 = Somewhat agree, 3 = Somewhat disagree, 4 = Totally disagree |
| | skole2 | My teachers care about me | 1 = Totally agree, 2 = Somewhat agree, 3 = Somewhat disagree, 4 = Totally disagree |
| | skole4 | I´m bored at school | 1 = Totally disagree, 2 = Somewhat disagree, 3 = Somewhat agree, 4 = Totally agree |
| | skole6 | I often don´t want to go to school | 1 = Totally disagree, 2 = Somewhat disagree, 3 = Somewhat agree, 4 = Totally agree |
| Local area | | Think about the areas around where you live. How good do you think that services for young people are in terms of: | |
| | naer1 | sports facilities | 1 = Very good, 2 = Quite good, 3 = Neither good nor bad, 4 = Quite bad, 5 = I don´t like it at all |
| | naer2 | culture (cinemas, concert veneus, libraries, etc.) | 1 = Very good, 2 = Quite good, 3 = Neither good nor bad, 4 = Quite bad, 5 = I don´t like it at all |
| Local safety | Neart | When you are out in the evening, do you feel safe in the local area where you live? | 1 = Yes, very safe, 2 = Yes, quite safe, 3 = Not sure, 4 = No, I don´t feel safe |
| Knowing drunk drivers | Kalko | Do you know young people who have driven a moped or other motorized vehicle after drinking alcohol? | 1 = No, 2 = Don´t know, 3 = Yes |
| **Other Risk Behaviors** | | | |
| Tobacco and nicotine product use | broeyk1 | Do you smoke? | 1 = Never, 2 = I used to, but I´ve stopped completely now, 3 = Less than once a week, 4 = Every week, but not every day, 5 = Every day |
| | broeyk2 | Do you use snus (tobacco that you put under your lip)? | 1 = Never, 2 = I used to, but I´ve stopped completely now, 3 = Less than once a week, 4 = Every week, but not every day, 5 = Every day |
| | broeyk3 | Do you use e-cigarettes/vape? | 1 = Never, 2 = I used to, but I´ve stopped completely now, 3 = Less than once a week, 4 = Every week, but not every day, 5 = Every day |
| Drug use | bdrug1 | used hash/maijuana/cannabis | 1 = Never, 2 = Once, 3 = 2–5 times, 4 = 6–10 times, 5 = 11 or more |
| | bdrug2 | used other drugs | 1 = Never, 2 = Once, 3 = 2–5 times, 4 = 6–10 times, 5 = 11 or more |
| Bullying others | Bmobb | Do you sometimes take part in teasing, threatening og freezing out other young people at school or in your free time? | 1 = Never, 2 = Almost never, 3 = Yes, around once a month, 4 = Yes, around once a fortnight, 5 = Yes, around once a week, 6 = Yes, several times a week |
| Aggressive behavior | | How many times have you done any of the following things over the past year (the past 12 months)? | |
| | bantiv1 | deliberately damaged or broken window panes, bus seats, post boxes, etc. (carried out vandalism) | 1 = Never, 2 = Once, 3 = 2–5 times, 4 = 6–10 times, 5 = 11 or more |
| | bantiv2 | ilegally spray-painted or tagged walls, buildings, trains, buses, etc. | 1 = Never, 2 = Once, 3 = 2–5 times, 4 = 6–10 times, 5 = 11 or more |
| | bantiv3 | Carried a knife or other weapons in places where it is not allowed. | 1 = Never, 2 = Once, 3 = 2–5 times, 4 = 6–10 times, 5 = 11 or more |
| | bantiv4 | Hacked, scammed someone, or engaged in other online criminal activities. | 1 = Never, 2 = Once, 3 = 2–5 times, 4 = 6–10 times, 5 = 11 or more |
| Alcohol use | balko1 | Do you ever drink any kinds of alcoholic drinks? | 1 = Never, 2 = I´ve just tried tasting them a few times, 3 = Occasionally, but less than once a month, 4 = Generally 1–3 times a month, 5 = Every week |
| | balko2 | How old were you the first time you drank so much alcohol? | 1 = 18 years or older, 2 = 17 years, 3 = 16 years, 4 = 15 years, 5 = 14 years, 6 = 13 years, 7 = Under 13 years |
| | balko3 | How many times have you had so much to drink that you felt clearly intoxicated over the past year (the past 12 months)? | 1 = Never, 2 = Once, 3 = 2–5 times, 4 = 6–10 times, 5 = 11 or more |

*(Continued)*

**Table 1.** (Continued)

| Composite variable | Measured variable | Questions | Response options (coding scheme) |
|---|---|---|---|
| Anti-social behavior | | How many times have you done any of the following things over the past year (the past 12 months)? | |
| | banti1 | take something from a shop without paying | 1 = Never, 2 = Once, 3 = 2–5 times, 4 = 6–10 times, 5 = 11 or more |
| | banti2 | not paid for the cinema, sporting events, bus and train tickets, etc. when you sholud have paid | 1 = Never, 2 = Once, 3 = 2–5 times, 4 = 6–10 times, 5 = 11 or more |
| | banti3 | spent the whole night away from home without your parents knowing where you were | 1 = Never, 2 = Once, 3 = 2–5 times, 4 = 6–10 times, 5 = 11 or more |
| | banti4 | skipped school | 1 = Never, 2 = Once, 3 = 2–5 times, 4 = 6–10 times, 5 = 11 or more |
| | banti5 | Consciously cheated on a test or assignment that was graded | 1 = Never, 2 = Once, 3 = 2–5 times, 4 = 6–10 times, 5 = 11 or more |

**Structure of the model.** The developed model for the PLS-SEM analysis is structured around the distinction between endogenous and exogenous latent variables, aligning with standard practices in structural equation modeling [30]. PLS-SEM uses the term "endogenous variables" instead of dependent variable and the term "exogenous variable" for independent variable [30]. This structure allows for a comprehensive examination of how various personal, social, and environmental factors influence adolescents' alcohol use and antisocial behaviors.

The principal endogenous variables for the PLS-SEM analysis were "alcohol use" and "antisocial behaviors." Other latent variables were treated as exogenous factors. Individual-level variables such as "socioeconomic status," "religion," "future outlook," and "depressive thoughts" capture personal characteristics that may influence alcohol use or antisocial behavior. The model also considers how adolescents spend their leisure time, including "physical activity," "gaming," and "social media use." Additionally, negative experiences such as "being bullied" and "sexual harassment" are included due to their potential role in increasing the likelihood of alcohol use or antisocial behavior. "COVID-19" is incorporated to reflect the unique impact of the pandemic on adolescents' daily lives, particularly regarding increased isolation.

Environmental factors such as "parental monitoring," "poor relationships with parents," "friendship support," "spending evenings with friends," "liking school," "local area," and "local safety" provide insight into the social contexts that shape adolescent behavior. The variable "knowing drunk drivers" is included to capture adolescents' indirect exposure to risky behavior within their peer environment, which may influence perceived norms and the acceptability of alcohol use.

Finally, risk behaviors including "tobacco and nicotine product use," "drug use," "bullying others," and "aggressive behavior" are included to examine co-occurring patterns that may contribute to alcohol use and antisocial outcomes.

**Data analysis.** PLS-SEM analysis was performed to investigate the associations between exogenous and endogenous variables. As the first step of the PLS-SEM analysis, model evaluation was conducted for both the measurement and structural models. Assessing the measurement models included evaluating factor loadings, reliability, convergent validity, discriminant validity, and indicator collinearity.

Factor loadings indicate the variance explained by a measured variable on a specific factor, with factor loadings of 0.5 or higher considered acceptable. Internal consistency reliability was evaluated using composite reliability (CR) for each latent variable. While Cronbach's alpha is frequently used for assessing internal consistency, it is known to be conservative and assumes equal indicator loadings across the population. Composite reliability, on the other hand, offers more flexibility regarding loading assumptions and is thus more suitable for PLS-SEM analysis [30], CR values of 0.7 or higher were considered acceptable.

The average variance extracted (AVE) was employed to evaluate the convergent validity of the latent variables, with a threshold of 0.4 or higher indicating adequate convergence [30]. Convergent validity ensures that indicators of a specific

construct are highly correlated, confirming that they accurately represent the same underlying concept [30]. For the structural models, collinearity among the predictor constructs was examined using the variance inflation factor (VIF). Although VIF values of 5.0 or higher suggest collinearity issues, a more conservative threshold of 3.3 was adopted [38].

Subsequently, PLS-SEM regression analysis was conducted to examine the relationships between exogenous and endogenous variables. To ensure the robustness of parameter estimates, bootstrapping with 10,000 resamples was employed to generate confidence intervals [30,39]. A stepwise regression method was employed, wherein the initial model included all potential explanatory variables, and non-statistically significant factors were progressively removed.

Prior to conducting stepwise regression, the sample was divided into two groups based on gender (boys and girls) to allow separate regression analyses for each group. The path coefficients for each group were then compared using multi-group analysis in PLS-SEM (PLS-MGA). Results indicated that 91.3% (42 out of 46 paths) of the path coefficients showed no statistically significant differences between the two groups. This result suggests that PLS-SEM analysis can be applied across genders [39,40].

The present study used SPSS Statistics 27 (IBM Corp., Armonk, N.Y., USA) to perform descriptive statistics, while R 4.4.1 (The R Foundation for Statistical Computing, Vienna, Austria) with the SEMinR package (version 2.3.3) was used for PLS-SEM analysis. The significance threshold was set at $p < 0.05$.

## Ethics

This study used the anonymized databases from Ungdata survey that has been approved by Norwegian Centre for Research Data (NSD). The survey was conducted in accordance with the Declaration of Helsinki. Participation was voluntary and informed consent was obtained from all students prior to the survey. Ethical issues were thus addressed by the institutions conducting the surveys and a separate ethics approval for this study was not necessary.

## Results

### Evaluation of measurement models

Table 2 presents the values for assessing the measurement model in Partial Least Squares Structural Equation Modeling (PLS-SEM). The values of factor loadings were 0.50 or higher, indicating the enough strength of relationships between measured variables and their corresponding latent variables [30]. Although some factor loadings were below 0.50, they were considered acceptable as their values were statistically significant [30]. Both composite reliability (CR), which assesses internal consistency reliability, and average variance extracted (AVE), which evaluates the validity of the latent variables, were within acceptable ranges, with CR being 0.70 or higher and AVE being 0.45 or higher. For the structural model assessment (Table 3 and 4), all variance inflation factor (VIF) values, which assess multicollinearity, were less than 3.30, indicating an acceptable range.

### Alcohol use

Table 3 presents the path coefficients and p-values for the structural model predicting alcohol use. The final model indicated that higher levels of the following factors were associated with an increased risk of alcohol use: non-religiousness (β = 0.129), depressive symptoms (β = 0.071), negative experiences related to COVID-19 (β = 0.088), experiences of sexual harassment (β = 0.111), frequent social media use (β = 0.071), spending evenings with friends (β = 0.072), knowing individuals who drive under the influence of alcohol (β = 0.186), tobacco and nicotine product use (β = 0.298), drug use (β = 0.067), and engagement in antisocial behaviors (β = 0.161). In contrast, some factors were associated with a reduced risk of alcohol use. These included a more negative future outlook (β = −0.046), greater time spent on gaming (β = −0.049), and lower levels of friendship support (β = −0.119).

**Table 2. Assessment of the measurement model.**

| Category | Composite variables | Measured variables | Mean | loading | rhoC | AVE |
|---|---|---|---|---|---|---|
| Outcome | Alcohol use | balko1 | 2.768 | 0.879 | 0.913 | 0.778 |
| | | balko2 | 3.360 | 0.827 | | |
| | | balko3 | 2.555 | 0.934 | | |
| | Anti-social behavior | banti1 | 1.166 | 0.703 | 0.771 | 0.490 |
| | | banti2 | 1.579 | 0.553 | | |
| | | banti3 | 1.701 | 0.798 | | |
| | | banti4 | 1.876 | 0.578 | | |
| | | banti5 | 1.591 | 0.505 | | |
| Psycosocial Background | Socioeconomic status | soes | 1.844 | 1.000 | 1.000 | 1.000 |
| | Religion | relig2 | 3.497 | 1.000 | 1.000 | 1.000 |
| | future outlook | fram1 | 1.370 | 0.636 | 0.769 | 0.633 |
| | | fram2 | 1.075 | 0.918 | | |
| | Depressionive symptom | depr1 | 2.398 | 0.774 | 0.882 | 0.600 |
| | | depr2 | 2.201 | 0.826 | | |
| | | depr3 | 2.166 | 0.773 | | |
| | | depr4 | 2.074 | 0.751 | | |
| | | depr5 | 2.168 | 0.684 | | |
| Adverse Experiences | Covid-19 | covid1 | 2.581 | 0.829 | 0.760 | 0.520 |
| | | covid2 | 1.915 | 0.576 | | |
| | | covid3 | 1.381 | 0.715 | | |
| | Being bullied | mobb1 | 1.518 | 0.547 | 0.769 | 0.462 |
| | | mobb2 | 1.332 | 0.593 | | |
| | | mobb3 | 1.279 | 0.852 | | |
| | | mobb4 | 1.053 | 0.648 | | |
| | Sexual harassment | sextrak1 | 1.290 | 0.735 | 0.835 | 0.561 |
| | | sextrak2 | 1.396 | 0.770 | | |
| | | sextrak3 | 1.347 | 0.817 | | |
| | | sextrak4 | 1.097 | 0.649 | | |
| Leisure Time Activities | Gaming | digi | 2.839 | 1.000 | 1.000 | 1.000 |
| | Social media use | sms1 | 4.826 | 1.000 | 1.000 | 1.000 |
| | Physical activity | fysak1 | 3.990 | 1.000 | 1.000 | 1.000 |
| Environmental Factors | Parental monitoring | fam1 | 1.674 | 0.615 | 0.817 | 0.537 |
| | | fam2 | 1.625 | 0.943 | | |
| | | fam3 | 1.680 | 0.728 | | |
| | | fam4 | 1.633 | 0.564 | | |
| | Poor relationships with parents | rfam1 | 1.991 | 0.879 | 0.767 | 0.627 |
| | | rfam2 | 1.706 | 0.688 | | |
| | Friendship support | venn1 | 1.569 | 0.517 | 0.773 | 0.549 |
| | | venn2 | 1.674 | 0.887 | | |
| | | venn3 | 1.443 | 0.556 | | |
| | Spending evenings with friends | bvenn | 2.183 | 1.000 | 1.000 | 1.000 |
| | Liking school | skole1 | 1.719 | 0.674 | 0.796 | 0.497 |
| | | skole2 | 1.866 | 0.814 | | |
| | | skole3 | 2.868 | 0.560 | | |
| | | skole4 | 1.915 | 0.678 | | |
| | Local area | naer1 | 2.392 | 0.895 | 0.886 | 0.795 |
| | | naer2 | 2.665 | 0.828 | | |
| | Local safety | naert | 1.508 | 1.000 | 1.000 | 1.000 |
| | Knowing drunk drivers | kalko | 1.651 | 1.000 | 1.000 | 1.000 |

*(Continued)*

**Table 2.** (Continued)

| Category | Composite variables | Measured variables | Mean | loading | rhoC | AVE |
|---|---|---|---|---|---|---|
| Other Risk Behaviors | Tobacco and nicotine product use | broeyk1 | 1.631 | 0.908 | 0.855 | 0.666 |
| | | broeyk2 | 1.803 | 0.817 | | |
| | | broeyk3 | 1.319 | 0.707 | | |
| | Drug use | bdrug1 | 1.159 | 0.985 | 0.878 | 0.784 |
| | | bdrug2 | 1.086 | 0.763 | | |
| | Bullying others | bmobb | 1.295 | 1.000 | 1.000 | 1.000 |
| | Aggressive behavior | bantiv1 | 1.118 | 0.912 | 0.817 | 0.537 |
| | | bantiv2 | 1.062 | 0.778 | | |
| | | bantiv3 | 1.109 | 0.576 | | |
| | | bantiv4 | 1.057 | 0.588 | | |

Notes:

rhoC: Composite reliability used to assess internal consistency reliability. Values of 0.7 or higher were considered acceptable.

AVE: Average variance extracted, used to assess the validity of the latent variables, with a threshold value of 0.45 or higher.

**Table 3. Path coefficients (β) for the structural model: alcohol use.**

| Category | Composite variables | Initial model | | Final model | |
|---|---|---|---|---|---|
| | | VIF | B | VIF | B |
| Psychosocial Background | Socioeconomic status | 1.101 | 0.014 | – | – |
| | **Religion** | 1.042 | **0.122\*\*\*** | 1.014 | **0.129\*\*\*** |
| | **Future outlook** | 1.305 | **−0.062\*\*** | 1.241 | **−0.046\*** |
| | **Depressionive symptom** | 1.489 | **0.061\*\*** | 1.291 | **0.071\*\*\*** |
| Adverse Experiences | **Covid-19** | 1.201 | **0.074\*\*\*** | 1.142 | **0.088\*\*\*** |
| | Being bullied | 1.832 | −0.022 | – | – |
| | **Sexual harassment** | 1.529 | **0.125\*\*\*** | 1.321 | **0.111\*\*\*** |
| Leisure Time Activities | **Gaming** | 1.204 | **−0.055\*\*** | 1.130 | **−0.049\*\*** |
| | **Social media use** | 1.160 | **0.069\*\*\*** | 1.135 | **0.071\*\*\*** |
| | Physical activity | 1.100 | 0.032 | – | – |
| Environmental Factors | Parental monitoring | 1.302 | 0.031 | – | – |
| | Poor relationships with parents | 1.253 | −0.015 | – | – |
| | **Friendship support** | 1.377 | **−0.116\*\*\*** | 1.266 | **−0.119\*\*\*** |
| | **Spending evenings with friends** | 1.302 | **0.073\*\*\*** | 1.295 | **0.072\*\*\*** |
| | Liking school | 1.344 | −0.002 | – | – |
| | Local area | 1.100 | 0.034 | – | – |
| | Local safety | 1.127 | −0.019 | – | – |
| | **Knowing drunk drivers** | 1.148 | **0.187\*\*\*** | 1.134 | **0.186\*\*\*** |
| Other Risk Behaviors | **Tobacco and nicotine product use** | 1.490 | **0.297\*\*\*** | 1.430 | **0.298\*\*\*** |
| | **Drug use** | 1.314 | **0.077\*\*\*** | 1.226 | **0.067\*\*\*** |
| | Bullying others | 1.347 | 0.006 | – | – |
| | Aggressive behavior | 1.763 | −0.029 | – | – |
| | **Anti-social behavior** | 1.996 | **0.171\*\*\*** | 1.495 | **0.161\*\*\*** |

**Table 4. Path coefficients (β) for the structural model: antisocial behavior.**

| Category | Composite variables | Initial model | | Final model | |
|---|---|---|---|---|---|
| | | VIF | B | VIF | B |
| Psychosocial Background | Socioeconomic status | 1.102 | −0.003 | – | – |
| | Religion | 1.068 | −0.023 | – | – |
| | Future outlook | 1.310 | 0.033 | – | – |
| | Depressionive symptom | 1.492 | 0.040 | – | – |
| Adverse Experiences | Covid-19 | 1.210 | −0.022 | – | – |
| | Being bullied | 1.833 | −0.015 | – | – |
| | **Sexual harassment** | 1.529 | **0.117***** | 1.294 | **0.111***** |
| Leisure Time Activities | **Gaming** | 1.205 | **−0.040*** | 1.133 | **−0.036*** |
| | Social media use | 1.169 | 0.007 | – | – |
| | Physical activity | 1.100 | 0.031 | – | – |
| Environmental Factors | **Parental monitoring** | 1.252 | **0.157***** | 1.219 | **0.085***** |
| | **Poor relationships with parents** | 1.241 | **0.076***** | 1.220 | **0.047***** |
| | Friendship support | 1.400 | −0.034 | – | – |
| | **Spending evenings with friends** | 1.300 | **0.074***** | 1.131 | **0.122***** |
| | **Liking school** | 1.335 | **0.065**** | 1.108 | **0.158***** |
| | Local area | 1.101 | −0.020 | – | – |
| | Local safety | 1.127 | −0.015 | – | – |
| | **Knowing drunk drivers** | 1.206 | **0.049***** | 1.186 | **0.077***** |
| Other Risk Behaviors | **Tobacco and nicotine product use** | 1.624 | **0.105***** | 1.507 | **0.066**** |
| | Drug use | 1.322 | 0.032 | – | – |
| | **Bullying others** | 1.324 | **0.105***** | 1.179 | **0.365***** |
| | **Aggressive behavior** | 1.507 | **0.355***** | 1.240 | **0.099***** |
| | **Alcohol use** | 1.755 | **0.149***** | 1.606 | **0.156***** |

Notes: * p < 0.5, ** p < 0.01, *** p < 0.001

VIF: The variance inflation factor, used to examine collinearity among the predictor constructs in the structural model. VIF values of 3.3 or higher indicate collinearity issues in this study.

-: Not included in the final model because of $p > .05$ in the first model.

## Antisocial behaviors

Table 4 presents the path coefficients and p-values for the structural model of antisocial behaviors. Several factors were associated with an increased likelihood of engaging in antisocial behaviors. These included higher levels of sexual harassment experiences (β = 0.111), greater parental monitoring (β = 0.085), poorer relationships with parents (β = 0.047), more frequent evenings spent with friends (β = 0.122), stronger liking of school (β = 0.158), knowing individuals who drive under the influence of alcohol (β = 0.077), use of tobacco or nicotine products (β = 0.066), bullying others (β = 0.365), aggressive behaviors (β = 0.099), and alcohol use (β = 0.156). In contrast, greater time spent gaming was associated with a lower likelihood of antisocial behaviors (β = −0.036).

Overall, the structural models identified several significant predictors of alcohol use and antisocial behaviors. For alcohol use, 13 out of 23 hypothesized paths (56.5%) were statistically significant, with risk increasing in association with factors such as religiousness, depressive symptoms, negative COVID-19 experiences, and various behavioral and social risk indicators, while protective effects were observed for positive future outlook, gaming, and friendship support. For antisocial behaviors, 11 out of 23 paths (47.8%) were significant, with increased risk linked to experiences such as sexual

harassment, poor parental relationships, peer and behavioral factors, and substance use, whereas gaming was associated with a reduced likelihood of antisocial behavior.

## Discussion

This study examined how individual and environmental factors—including influences from family, peers, school, and the local community—are associated with alcohol use and antisocial behaviors among high school students in Finnmark, using Partial Least Squares Structural Equation Modeling (PLS-SEM). The analysis identified several significant predictors. Alcohol use was positively associated with factors such as religiousness, depressive symptoms, adverse COVID-19 experiences, sexual harassment, social media use, peer interactions, substance use, and antisocial behaviors, while a positive future outlook, gaming, and friendship support were protective. Antisocial behaviors were linked to a similar set of risk factors, including parental monitoring, poor parental relationships, and alcohol use, whereas gaming emerged as a protective factor.

### Psychosocial background

Results showed higher levels of non-religiousness and depressive symptoms were associated with increased alcohol use, while a more negative future outlook was linked to a reduced risk of alcohol use. Socioeconomic status (SES) showed no association with either alcohol use or antisocial behaviors. None of the psychosocial background factors were associated with antisocial behaviors.

The findings regarding the associations between these factors and alcohol use were both consistent with and contradictory to prior research. Studies have reported that higher levels of depressive symptoms are linked to an earlier onset of alcohol use, more frequent consumption, and intoxication [9]. Similarly, mental health status was found to be significantly associated with binge drinking among adolescents in the Oslo metropolitan area [8]. Furthermore, religious importance has been identified as a protective factor against drinking behaviors among adolescents in Northern Norway [14] as well as binge drinking among adolescents in the Oslo metropolitan area [8]. In contrast, a study in Vestland County, located on Norway's western coast, identified parental socioeconomic status, depression, and religiosity as factors linked to antisocial behaviors among adolescents [16]. However, in our study, no individual factors—including these three—showed any association with antisocial behaviors. This discrepancy could stem from regional differences, cultural variations, or differences in the study populations.

SES did not show any association with alcohol use or antisocial behaviors in our study. While many studies suggest that household SES influences adolescent risk behaviors, some research in high-income countries has reported different trends, likely due to the relatively small income differences among households [3,4,41]. For example, studies in high-income countries have suggested that the impact of SES on adolescent behaviors may be diminished or may increase alcohol consumption due to easier access to alcohol [42,43]. This could explain why, in Finnmark, other factors—such as social influences, personality traits, or community norms—may play a more significant role in shaping adolescents' engagement in alcohol use and antisocial behaviors.

An increased risk of alcohol use is associated with a more positive future outlook. One possible explanation is that adolescents with a strong future orientation may engage more in social activities where alcohol consumption is part of social bonding. This aligns with the current study's finding that time spent with friends in the evening is linked to increased alcohol use. Furthermore, regional factors in Finnmark, such as limited recreational activities and social isolation [2,44], may influence drinking patterns. Adolescents with a strong future outlook may engage in drinking as a way to integrate into social groups or to combat boredom in a remote region.

Multi-group analysis in PLS-SEM analysis revealed no significant gender differences in the associations between protective and risk factors and adolescent alcohol use or antisocial behaviors in Finnmark. This suggests that both male and female students are similarly influenced by environmental and social factors. These findings align with research showing

minimal gender differences in drinking patterns among Indigenous Sámi and non-Indigenous youth in Northern Norway [13]. However, studies in Oslo and western Norway have linked gender, along with other factors like parental education and religiosity, to binge drinking and antisocial behaviors [8,16]. The lack of gender differences in Finnmark may reflect regional or cultural influences, highlighting the need for further research across different settings.

### Adverse experiences

Results showed frequent experience of sexual harassment was linked to increased risk of both alcohol use and antisocial behaviors, while being bullied showed no significant relationship with either. These findings align with research indicating that hate speech and sexual harassment predict antisocial behaviors [16] and that sexual harassment victimization is linked to substance use as a coping mechanism [45]. Although being bullied is not a predictor of increased alcohol use or antisocial behaviors, the higher prevalence of bullying in Finnmark compared to the national average [2] remains a concern. Previous studies have also found that Sámi youth in Northern Norway experience higher levels of discrimination [37,46]. Given that Norway's Education Act § 9a-3 mandates a safe school environment [47], addressing bullying, sexual harassment, and hate speech remains crucial, particularly for high school students [48].

This study showed that COVID-19-related negative experiences among adolescents in Finnmark were linked to an increased risk of alcohol use but not antisocial behaviors. Previous research showed mixed results. Some studies reported increased drinking, especially among males and younger students, due to stress and reduced social contact [49,50]. Others found overall alcohol use declined, except among high-risk individuals facing mental health challenges [51]. The findings of current study align with studies linking pandemic stress to alcohol use but differ from those showing reduced consumption.

### Leisure time activities

The results indicate that increased frequency of social media use was associated with a higher risk of alcohol use, but not with antisocial behaviors. In contrast, greater time spent on gaming showed negative associations with both alcohol use and antisocial behaviors, suggesting that more time spent gaming was linked to reducing the risk of alcohol use and fewer antisocial behaviors.

Greater time spent on gaming may indicate a displacement of offline social interactions, as adolescents who engage extensively in online gaming may have fewer opportunities for in-person peer activities, which are often associated with increased risk behaviors. Prior research has shown that spending time with friends in the evening is linked to higher levels of alcohol use and antisocial behaviors [7,18]. A recent study using the same Ungdata dataset found that adolescents in Finnmark who spent more time with friends in the evening tended to spend less time gaming [52]. While this does not rule out the possibility of shared digital activities (e.g., gaming or watching videos together), the observed negative association between gaming and risk behaviors may reflect a broader shift away from in-person socializing.

Social media use, on the other hand, was linked to a higher risk of alcohol use. This finding aligns with research showing that exposure to drinking-related content on social platforms can normalize alcohol consumption. Descriptive norms—perceptions of peers' drinking behaviors—have been found to influence adolescent alcohol use [53,54]. These findings underscore the importance of examining gaming and social media use separately [52], as they are linked to distinct mechanisms. Social media use, for instance, is more closely associated with peer dynamics and negative experiences, including depressive symptoms and sexual harassment. Furthermore, broader trends such as increasing screen time among Norwegian adolescents [55] and concerns about the impact of problematic social media use on mental health [56] emphasize the need for further research into how digital engagement shapes adolescent risk behaviors.

Physical activity was not associated with alcohol use or antisocial behaviors, contrasting with some research showing that structured activities, including sports, can serve as protective factors [18]. However, other studies indicate that sports participation can also be linked to alcohol consumption due to team bonding and celebratory drinking cultures

[3]. Regional characteristics in Finnmark—such as long travel distances and a more dispersed population—may contribute to a lower participation in organized physical activities among adolescents in Finnmark than the national average [2]. These factors could help explain why physical activity did not show the expected protective associations in this context.

## Environmental factors

Findings revealed that spending evenings with friends in the evening was significantly associated with both alcohol use and antisocial behaviors. This suggests that unstructured socializing with peers may provide opportunities for risk behaviors, including alcohol use and antisocial behaviors among students in Finnmark. A ten-year trend analysis showed that adolescents in Finnmark participate less frequently in organized leisure activities than the national average [2], which may contribute to increased unsupervised time. These results are consistent with previous studies linking heavy episodic drinking to evening socializing with peers and low parental control [7]. Similarly, increased antisocial behaviors, such as physical fighting among adolescents in Oslo, were linked to spending more time with friends during the evenings [18].

Results on association with friendship support and alcohol use indicated that adolescents with stronger peer connections may be more likely to engage in drinking. This association may reflect social norms within peer groups where alcohol use is accepted or encouraged. Although peer support is generally considered protective against risk behaviors [57], Social Learning Theory [58] suggests that behavior is shaped through observation and interaction. In peer networks where alcohol consumption is common, adolescents may be more inclined to adopt similar behaviors.

Several studies support this idea, showing that adolescents with strong peer connections—especially in environments where drinking is normalized—are more likely to consume alcohol. For instance, peers' drinking behavior has consistently been associated with adolescents' own alcohol use [53,54]. One study found that this association helped explain the link between drinking with peers and individual alcohol use for both males and females, with a somewhat stronger effect among males [59]. Additionally, perceived peer approval of drinking has been shown to influence both current and future drinking behavior, underscoring the role of social conformity in shaping adolescent substance use [60].

In contrast, family relationships and linking school were associated only with antisocial behaviors, suggesting that while these environments contribute to behavioral regulation, they may not directly influence alcohol consumption. These findings contrast with research from other Norwegian regions. For example, in Vestland County, family relationships and school well-being were significantly associated with antisocial behaviors among high school students [16]. In contrast, studies in the Oslo metropolitan area indicated that parent–adolescent relationships and parental control were significantly associated with binge drinking, as well as cannabis, and tobacco use [8]. Although the measures and study designs differ from those used in the present study, these findings suggest potential regional variation in the influence of family and school environments on adolescent behavior.

Unlike early adolescence, high school-aged adolescents (16–19 years old) exhibit increased autonomy and are more influenced by external factors, particularly peers at school and in their social circles [3]. This study indicates that, unlike high school students in the Oslo metropolitan area, those in Finnmark appear to be more influenced by their peers than by family or school environments. One possible explanation for this difference lies in variations in parental attitudes and cultural expectations. For example, the traditional indigenous Sámi family structure in the Sápmi region, including Finnmark, differs from the dominant Norwegian family model, emphasizing collective responsibility while fostering autonomy [61,62]. Sámi child-rearing practices prioritize independence, resilience, and indirect control methods, blending autonomy with collectivist values [63,64]. In contrast, mainstream Norwegian parenting tends to involve more direct supervision and structured guidance, particularly regarding adolescent behavior. These differing parental attitudes may shape how adolescents in Finnmark engage with peers and navigate risk behaviors. Future research should further examine how such cultural parenting practices influence adolescent decision-making in both indigenous and non-indigenous communities.

## Other risk behaviors

The results showed that several risk behaviors—such as increased use of tobacco or nicotine products, drug use, bullying others, and aggressive behavior—were associated with a higher likelihood of both alcohol use and antisocial behaviors. Similarly, alcohol use predicted greater engagement in antisocial behavior, and vice versa. These findings are consistent with Problem Behavior Theory [65] and developmental cascade models [66], which propose that risk behaviors tend to co-occur and reinforce each other over time. The results also align with previous research from Norway indicating that daily smoking is associated with increased levels of heavy episodic drinking [7] and antisocial behavior [16].

## Theoretical implications

This study deepens the understanding of adolescent risk behaviors by emphasizing the strong influence of peer interactions, particularly unstructured socialization, on alcohol use and antisocial behaviors. This supports Social Learning Theory [58], which highlights that behaviors are learned through observation and reinforcement within social circles. They also align with Bronfenbrenner's ecological systems theory [22], demonstrating how various elements within the microsystem, especially peer relationships, shape adolescent behavior—often exerting greater influence than family or school environments. The limited association between family factors, school engagement, and alcohol use in Finnmark contrasts with prior research emphasizing parental control as a protective factor. This suggests that peer influences may override traditional protective mechanisms in certain regional contexts. The role of digital media in shaping alcohol consumption behaviors suggests the need to integrate traditional peer influence models with digital socialization theories [67]. The study also reinforces the cycle of violence hypothesis [68] by linking sexual harassment experiences to risk behaviors. Furthermore, the results on notable pattern of co-occurring risk behaviors are consistent with Problem Behavior Theory [65] and developmental cascade models [66], which argue that risk behaviors often reinforce each other over time, contributing to a cycle of escalating risk.

## Political implications

The findings highlight the need for policies that mitigate adolescent risk behaviors through community-based interventions. Given the strong influence of peer interactions, structured after-school activities and supervised youth programs could reduce unstructured socialization and promote positive peer engagement. Although family and school environments played a weaker role in this study, they remain critical, and strengthening family-school partnerships—particularly in culturally diverse regions like Finnmark—could reinforce protective factors. Recognizing the distinct associations between social media use and gaming is important for policy development. While increased social media use was linked to a higher risk of alcohol use, gaming showed a negative association, possibly reflecting reduced offline socializing where risk behaviors often occur. These findings highlight the need for targeted strategies that address different types of digital engagement in adolescent health prevention efforts. Experiences of sexual harassment are linked to higher risks of alcohol use and antisocial behaviors, and bullying others also contributes to antisocial behavior. Comprehensive policies aimed at reducing bullying and harassment could help mitigate these risks. Since tobacco use, drug use, bullying, and aggression are linked to both alcohol use and antisocial behaviors, intervention strategies should address this broader constellation of risk behaviors and target them together.

While psychological factors showed weaker associations with risk behaviors, accessible mental health services remain essential for adolescents dealing with stressors like bullying and discrimination. Addressing adolescent alcohol use and antisocial behaviors requires a comprehensive, multi-faceted approach that integrates peer, family, school, and community interventions.

## Limitation

The present study had several limitations that should considered. First, the Ungdata survey is school based, which excludes students who have dropped out of school. Second, the response rate of high school students was lower than

that of junior high students, with only around 66–68% of high school students responding, compared to 80–83% of junior high students. This lower response rate may have introduced some bias in the results.

Additionally, the survey used cross-sectional data, preventing the establishment of causal inferences or temporal relationships between the studied variables. While significant associations were observed, it is not possible to determine the directionality of these relationships. For instance, the study identified correlations between peer interactions and substance use, but it remains unclear whether spending time with peers increases alcohol consumption or whether adolescents who consume alcohol are more likely to seek peer interactions. Additionally, some questions were answered retrospectively, making them susceptible to recall bias and memory lapses.

Another limitation is the inconsistency in the time frames used to assess variables. Free time was measured over the past 7 days, media use had no specific time frame, antisocial behavior and drug use were assessed over the past year, and depressive symptoms was measured over the past week. The timing of data collection, such as during exams or emotionally stressful periods, may have influenced results, especially for variables like depressive symptoms, which are assessed over shorter time frames.

The study did not explicitly account for age in its measurement or analysis. While the dataset spans three school grade levels—an age range that can represent significant developmental differences during adolescence—direct age data were not available in the Ungdata survey. As a proxy, school grade was used to approximate age; however, this approach does not account for intra-grade age variation, which could potentially influence several of the studied outcomes.

Lastly, the survey did not include indicators related to ethnicity. Given the distinct socio-cultural and multi-ethnic contexts of Finnmark County, particularly the presence of the Sámi and Kven people, as well as individuals from Northern Finland, Northern Sweden, and Russia, and the inter-marriage of these groups, the diversity within these populations may have influenced factors affecting adolescents, such as family, social networks, health, and risk behaviors. Different cultures and different group of people shape individuals' behavioral decisions in various ways. Future studies should include ethnicity-related factors, such as ethnic identity, to better understand these issues.

## Conclusion

This study examined the associations between individual, social, and environmental factors and adolescent alcohol use and antisocial behaviors in Finnmark County, utilizing Partial Least Squares Structural Equation Modeling. The findings indicate that peer-related variables—particularly unstructured evening socialization—are significantly associated with increased alcohol use and antisocial behavior. These results underscore the central role of peer environments in shaping adolescent risk behavior in this regional context.

Increased time spent on online gaming was associated with a reduced risk of both alcohol use and antisocial behaviors, suggesting that digital platforms may limit real-life peer interactions that typically foster risk behaviors. In contrast, social media use was associated with an increased risk of alcohol consumption, highlighting the significant influence of digital peer dynamics on adolescent drinking. Experiences of sexual harassment, as well as other concurrent risk behaviors such as smoking, drug use, bullying, and aggression, were consistently associated with increased levels of both alcohol use and antisocial behaviors. Family and school environments demonstrated a more limited association with alcohol use but were linked to antisocial behaviors. Socioeconomic status and gender were not significantly associated with either outcome, and psychosocial factors such as depressive symptoms and religiosity showed selective associations, primarily with alcohol use.

These findings suggest that preventive efforts in Finnmark should prioritize peer-focused strategies and address digital media exposure as key influences on adolescent risk behavior. Future research should incorporate ethnicity-related variables to improve the contextual understanding of these behaviors.

### Declaration of generative AI and AI-assisted technologies in the writing process

During the preparation of this work the author used ChatGPT-4o mini, in order to improve the readability and language of the manuscript, as well as to check grammar.

## Acknowledgments

The author would like to express sincere appreciation to Sami Norwegian National Advisory Unit for Mental Health and Substance Use, Finnmark Hospital Trust, for providing a supportive research environment.

## Author contributions

**Conceptualization:** Shiho Hansen.

**Data curation:** Shiho Hansen.

**Formal analysis:** Shiho Hansen.

**Funding acquisition:** Shiho Hansen.

**Investigation:** Shiho Hansen.

**Methodology:** Shiho Hansen.

**Project administration:** Shiho Hansen.

**Validation:** Shiho Hansen.

**Visualization:** Shiho Hansen.

**Writing – original draft:** Shiho Hansen.

**Writing – review & editing:** Shiho Hansen.

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
