## [Decision Letter · Decision Letter 0]

Dear Dr. Hansen,

Thank you for submitting your manuscript to PLOS ONE. After careful consideration, we feel that it has merit but does not fully meet PLOS ONE’s publication criteria as it currently stands. Therefore, we invite you to submit a revised version of the manuscript that addresses the points raised during the review process.

We look forward to receiving your revised manuscript.

Kind regards,

Md. Shiblur Rahaman, Ph.D.

Academic Editor

PLOS ONE

Journal Requirements:

“This research was partially supported by the internal research fund of Finnmark Hospital Trust. None of the funding providers had any role in the data analyses and interpretation, nor had they any right to approve or disapprove the writing and publication of the manuscript.”

“The author declares no conflicts of interest.”

Reviewers' comments:

Reviewer's Responses to Questions

**Comments to the Author**

1. Is the manuscript technically sound, and do the data support the conclusions?

Reviewer #1: Partly

Reviewer #2: Yes

2. Has the statistical analysis been performed appropriately and rigorously?

Reviewer #1: No

Reviewer #2: Yes

3. Have the authors made all data underlying the findings in their manuscript fully available?

Reviewer #1: Yes

Reviewer #2: No

4. Is the manuscript presented in an intelligible fashion and written in standard English?

Reviewer #1: Yes

Reviewer #2: Yes

Reviewer #1: Dear Editor of PLOS One,

the manuscript "Assessment of the influential mechanisms of adolescent risk-taking behaviors and

protective and risk factors among high school students in Finnmark, Arctic Norway" by Shiho Hansen is about an important topic as there is only little or no reserach on substance use and anti-social behaviours among Sami adolescents. However, the manuscript has many shortcomings which need to be solved before it can be accepted for publication.

To begin with, there is a lot of research in the associations of various peer, family and school related risk and protective factors related to anti-social behaviours. It should be more clerly stated, what's new about this study, why would Sami adolescents be an interesting group to study? Do you suspect that this group would be somehat different or have different living conditions from those in the rest of Norway?

It is interesting that the trends of substance use among adolescents look similar in Finnmark as in rest of Norway, but there are differences with regard to trends in anti-social behaviours (asb). Thus, to me it looks like it would be more interesting to study asb and variables associated to it and try to reflect the results against other Norwegian adolescents. Would it be possioble to get the whole Ungdata data set and compare Sami adolescents with others?

A technical problem is that there are too much both outcomes and explanatory variables (92 in total!). Choosing only either substance use or asb (both include several separate measures as outcomes). would make the analysis simpler and the ms easier to follow. In addition, as substance use and other risk behaviours have often been shown to be interrelated, there is no need to take all these variables in the analysis. I also wonder, why you need so many explanatory variables. For example, what is the reason for including the "covid" variable? To continue, variables listed in Table 1 don't seem to be "latent variables". Did you use LCA or some other way of constructing latent variables? . See for example:

Lanza, S. T., & Rhoades, B. L. (2013). Latent class analysis: an alternative perspective on subgroup analysis in prevention and treatment. Prevention science : the official journal of the Society for Prevention Research, 14(2), 157–168. https://doi.org/10.1007/s11121-011-0201-1.

To conclude, I suggest simplifying the manuscript, and comparing Sami adolescents against other adolescents in Norway. I also think more theoretical background is needed when rationalizing the research questions and hypothesis. In its present form the ms is more like a report of all that is included in Ungdata in a sample of Sami adolescents. If everything needs to be reported, I suggest writing 2-3 separate articles.

Reviewer #2: The manuscript addresses a pertinent topic and takes a holistic approach to understanding the individual and environmental factors (family, school, peers) that may contribute to risky behaviors (substance use, antisocial behavior) among Norwegian adolescents from the Finnmark region.

The data from "Ungdata 2021" were used, allowing for a considerable sample size (2,129), making the results more robust. Sophisticated statistical analyses (PLS-SEM) with regression were used in the data analysis, allowing for exploring complex relationships between variables.

The results indicate that spending time with friends in the evenings is strongly associated with alcohol and tobacco use and rule violations. Family relationships influence rule violations and bullying behaviors, while school-related factors are only associated with rule violations. Neither family relationships nor school-related factors appear to be associated with substance use (i.e., alcohol, tobacco, or drug use).

However, as the authors also acknowledge, the study did not explore sociocultural and ethnic factors, which may play an important role in the population and context of the Finnmark region. The manuscript would benefit from providing more concrete data to support the relevance of this study in the specific region of Finnmark (e.g., data on delinquent behavior and substance use). In addition, more detailed information on that region's sociocultural characteristics could help contextualize or better discuss some of the results. Some results require further discussion or justification, especially those that contradict the research or are considered "surprising" (e.g., religiosity associated with alcohol consumption).

Overall, the manuscript addresses a relevant topic with potential implications for policy and practice in the specific region where the study was conducted. However, the authors must overcome some conceptual, methodological, and interpretative limitations for its publication. In this context, the manuscript requires major revisions.

Important aspects to improve

- At the conceptual level, some constructs need to be clarified and better defined, namely "risk-taking behaviors," "protective factors," and "risk factors."

- Explicitly acknowledge and discuss some limitations: the fact that this is a cross-sectional study does not allow for establishing causal relationships. For example, the study identifies correlations between interactions with peers and substance use; it is not possible to determine whether spending time with peers leads to increased alcohol consumption or whether young people who consume alcohol tend to spend more time with peers.

- Several results contradict previous research or are paradoxical (e.g., religiosity positively associated with alcohol consumption, physical exercise positively associated with smoking). These results require more detailed explanations or justifications to avoid confusion. Review the statistical modeling process to ensure that these results are robust. If so, explanations based either on theory or on the specificities of the specific local context should be provided.

- Unlike substance use, attitudes related to antisocial behavior were not analyzed and are an important variable for understanding this type of behavior. It would be pertinent for the authors to consider the implications of this limitation when discussing some results.

- Several sections and many paragraphs do not present references to support some statements. For example, the section "Theoretical and Contextual Framework" only has one reference (about Bronfenbrenner's Social-Ecological Model), "Summary of studies in Norway"; the role of digital media in the formation of social norms. It is suggested to enrich the bibliographic review with relevant studies

Other issues:

- Tables 2 and 3 present dense information that could be simplified, making them easier to read. The authors could consider highlighting key coefficients and removing non-significant paths to make them easier to read.

- The presentation of p-values in tables 3a and 3b is not uniform. It is suggested that all significant p-values be presented with three decimal places to ensure uniformity.

- In the "data analysis" section, the authors report, "This result suggests that PLS-SEM analysis can be applied across (78 out of 82 paths) of the path coefficients showed no statistically significant differences genders." It could enrich the work by discussing/explaining that there are no gender differences in the discussion or conclusion sections.

**Do you want your identity to be public for this peer review?** For information about this choice, including consent withdrawal, please see our Privacy Policy

Reviewer #1: No

Reviewer #2: **Yes: ** Ana Rita Conde

---

## [Author Response · Author response to Decision Letter 1]

19 Mar 2025

Response to the Reviewers’ Comments and Concerns

Comments on Journal Requirements:

Response 1:

Thank you for your comment. I have revised the entire manuscript to ensure it meets PLOS ONE’s style requirements.

“This research was partially supported by the internal research fund of Finnmark Hospital Trust. None of the funding providers had any role in the data analyses and interpretation, nor had they any right to approve or disapprove the writing and publication of the manuscript.”

Response 2:

Thank you for your feedback. I have updated the Funding Statement as follows (page 33, lines 3-9):

“This research was supported by the internal research fund of Finnmark Hospital Trust. Additionally, the study was made possible through research time approved by my affiliated institution, the Sámi Norwegian National Advisory Unit for Mental Health and Substance Use (SANKS). None of the funding providers had any role in the data analyses and interpretation, nor had they any right to approve or disapprove the writing and publication of the manuscript. There was no additional external funding received for this study.”

“The author declares no conflicts of interest.”

Response 3:

Thank you for your comment. I have included the statement: “The author has declared that no competing interests exist” in both the Competing Interests (page 33, lines 10-11) and the cover letter.

Response 4:

Thank you for your feedback regarding the data availability statement. We have revised the statement to clarify the restrictions on sharing the data due to ethical and legal considerations. The dataset contains sensitive information and is subject to data protection regulations. Therefore, access is restricted and requires approval from the Norwegian Agency for Shared Services in Education and Research (Sikt) and Norwegian Social Research (NOVA).

For researchers interested in accessing the data, we have provided a link to Sikt’s Survey Bank, where data access requests can be submitted: https://surveybanken.sikt.no/no/study/NSD3157/1?file=9c72d168-67ae-4c15-82d9-738535771af9/4&type=studyMetadata.

(see page 33, lines 12-21)

Comments from the Reviewer #1:

The manuscript "Assessment of the influential mechanisms of adolescent risk-taking behaviors and

protective and risk factors among high school students in Finnmark, Arctic Norway" by Shiho Hansen is about an important topic as there is only little or no reserach on substance use and anti-social behaviours among Sami adolescents. However, the manuscript has many shortcomings which need to be solved before it can be accepted for publication.

To begin with, there is a lot of research in the associations of various peer, family and school related risk and protective factors related to anti-social behaviours. It should be more clerly stated, what's new about this study, why would Sami adolescents be an interesting group to study? Do you suspect that this group would be somehat different or have different living conditions from those in the rest of Norway?

It is interesting that the trends of substance use among adolescents look similar in Finnmark as in rest of Norway, but there are differences with regard to trends in anti-social behaviours (asb). Thus, to me it looks like it would be more interesting to study asb and variables associated to it and try to reflect the results against other Norwegian adolescents. Would it be possioble to get the whole Ungdata data set and compare Sami adolescents with others?

A technical problem is that there are too much both outcomes and explanatory variables (92 in total!). Choosing only either substance use or asb (both include several separate measures as outcomes). would make the analysis simpler and the ms easier to follow. In addition, as substance use and other risk behaviours have often been shown to be interrelated, there is no need to take all these variables in the analysis. I also wonder, why you need so many explanatory variables. For example, what is the reason for including the "covid" variable? To continue, variables listed in Table 1 don't seem to be "latent variables". Did you use LCA or some other way of constructing latent variables? . See for example:

Lanza, S. T., & Rhoades, B. L. (2013). Latent class analysis: an alternative perspective on subgroup analysis in prevention and treatment. Prevention science : the official journal of the Society for Prevention Research, 14(2), 157–168. https://doi.org/10.1007/s11121-011-0201-1.

To conclude, I suggest simplifying the manuscript, and comparing Sami adolescents against other adolescents in Norway. I also think more theoretical background is needed when rationalizing the research questions and hypothesis. In its present form the ms is more like a report of all that is included in Ungdata in a sample of Sami adolescents. If everything needs to be reported, I suggest writing 2-3 separate articles.

Response to Reviewer #1:

Thank you for your insightful and constructive feedback. I have made the following revisions based on your suggestions. Changes occurred with refinement of outcomes and explanatory variables have been applied throughout the manuscript.

1. Refinement of Outcomes and Explanatory Variables:

o The number of outcome and explanatory variables has been reduced to align with Bronfenbrenner's Social-Ecological Model, which underpins this study.

o The focus is now on alcohol use (substance use) and rule-breaking behaviors (antisocial behaviors) due to their higher prevalence and significant health implications. The distinction between alcohol use and antisocial behaviors is discussed in the Discussion section.

2. Explanatory Variables:

o Variables not central to the study, such as positive feelings and quality of life, have been removed due to their weak associations with the selected outcomes.

o The COVID-19 variable has been retained given its relevance, as data collection occurred during the pandemic (2021).

3. Comparison with National Data:

o A comparative analysis between adolescents in Finnmark and the national adolescent population is covered in a separate manuscript (insert title).

o To maintain focus and avoid redundancy, this study does not include such comparisons.

4. Theoretical Background:

o Background and Theoretical and Contextual Framework has been expanded to strengthen the rationale for the research questions (page 2, line 19 – page 8, line 8).

Comments from the Reviewer #2:

The manuscript addresses a pertinent topic and takes a holistic approach to understanding the individual and environmental factors (family, school, peers) that may contribute to risky behaviors (substance use, antisocial behavior) among Norwegian adolescents from the Finnmark region.

The data from "Ungdata 2021" were used, allowing for a considerable sample size (2,129), making the results more robust. Sophisticated statistical analyses (PLS-SEM) with regression were used in the data analysis, allowing for exploring complex relationships between variables.

The results indicate that spending time with friends in the evenings is strongly associated with alcohol and tobacco use and rule violations. Family relationships influence rule violations and bullying behaviors, while school-related factors are only associated with rule violations. Neither family relationships nor school-related factors appear to be associated with substance use (i.e., alcohol, tobacco, or drug use).

However, as the authors also acknowledge, the study did not explore sociocultural and ethnic factors, which may play an important role in the population and context of the Finnmark region. The manuscript would benefit from providing more concrete data to support the relevance of this study in the specific region of Finnmark (e.g., data on delinquent behavior and substance use). In addition, more detailed information on that region's sociocultural characteristics could help contextualize or better discuss some of the results. Some results require further discussion or justification, especially those that contradict the research or are considered "surprising" (e.g., religiosity associated with alcohol consumption).

Overall, the manuscript addresses a relevant topic with potential implications for policy and practice in the specific region where the study was conducted. However, the authors must overcome some conceptual, methodological, and interpretative limitations for its publication. In this context, the manuscript requires major revisions.

Important aspects to improve

- At the conceptual level, some constructs need to be clarified and better defined, namely "risk-taking behaviors," "protective factors," and "risk factors."

Response 1:

Thank you for your valuable feedback. To improve conceptual clarity, we have explicitly defined risk-taking behaviors, risk and protective factors within the Model Development and Structure of the Model (page 11, line 10 – page 15, line 14). These definitions are now aligned with the theoretical framework of the study.

- Explicitly acknowledge and discuss some limitations: the fact that this is a cross-sectional study does not allow for establishing causal relationships. For example, the study identifies correlations between interactions with peers and substance use; it is not possible to determine whether spending time with peers leads to increased alcohol consumption or whether young people who consume alcohol tend to spend more time with peers.

Response 2:

Thank you for pointing this out. We have expanded the Limitation section to explicitly state that, while the findings indicate associations between peer interactions and substance use, the directionality of these relationships remains uncertain. We now clarify that causality cannot be inferred, as spending time with peers could lead to increased alcohol consumption, or vice versa (page 31, lines 1-8).

- Several results contradict previous research or are paradoxical (e.g., religiosity positively associated with alcohol consumption, physical exercise positively associated with smoking). These results require more detailed explanations or justifications to avoid confusion. Review the statistical modeling process to ensure that these results are robust. If so, explanations based either on theory or on the specificities of the specific local context should be provided.

Response 3:

Thank you for your observation. I appreciate your valuable feedback. To address this, I have taken several steps:

• Under Methods, I have clarified the sample size determination to confirm the robustness of the analysis (Partial Least Squares Structural Equation Modeling and Sample Size, page 9 line 19 – page 10, line 25). In the Data Analysis section, I have explicitly discussed the reliability, validity, and multicollinearity assessments conducted prior to the PLS-SEM analysis (page 15, line 14 – page 16, line 9). Additionally, I have included a description of bootstrapping with 10,000 resamples to ensure the robustness of parameter estimates (page 16, lines 10-15).

• Discussion section has been revised (page 16 line 8 – page 29, line 6) to provide a more detailed analysis of the findings. This includes:

1. Inconsistencies with previous research, such as religiosity not acting as a protective factor against alcohol use, a positive future outlook being linked to increased drinking, and no significant gender differences in alcohol use or antisocial behaviors;

2. Negative associations observed between digital media use and risk behaviors, as well as strong peer relationships correlating with increased alcohol use;

3. No associations found between family and school influences on alcohol use, individual factors and antisocial behaviors, socioeconomic status and risk behaviors, being bullied and risk behaviors, physical activity and risk behaviors, or attitudes toward antisocial behaviors.

- Unlike substance use, attitudes related to antisocial behavior were not analyzed and are an important variable for understanding this type of behavior. It would be pertinent for the authors to consider the implications of this limitation when discussing some results.

Response 4:

Thank you for pointing this out. We acknowledge this limitation and have explicitly stated in the Limitation section that attitudes toward antisocial behaviors were not assessed. We now discuss how this omission limits our ability to examine how individual beliefs and perceived social norms influence such behaviors (page 31, lines 15-19).

- Several sections and many paragraphs do not present references to support some statements. For example, the section "Theoretical and Contextual Framework" only has one reference (about Bronfenbrenner's Social-Ecological Model), "Summary of studies in Norway"; the role of digital media in the formation of social norms. It is suggested to enrich the bibliographic review with relevant studies

Response 5:

Thank you for your c

---

## [Decision Letter · Decision Letter 1]

Dear Dr. Shiho,

Thank you for submitting your manuscript to PLOS ONE. After careful consideration, we feel that it has merit but does not fully meet PLOS ONE’s publication criteria as it currently stands. Therefore, we invite you to submit a revised version of the manuscript that addresses the points raised during the review process.

We look forward to receiving your revised manuscript.

Kind regards,

Jiankun Gong

Academic Editor

PLOS ONE

Journal Requirements:

Reviewers' comments:

Reviewer's Responses to Questions

**Comments to the Author**

Reviewer #1: (No Response)

Reviewer #2: All comments have been addressed

2. Is the manuscript technically sound, and do the data support the conclusions?

Reviewer #1: Partly

Reviewer #2: Yes

3. Has the statistical analysis been performed appropriately and rigorously?

Reviewer #1: Yes

Reviewer #2: Yes

4. Have the authors made all data underlying the findings in their manuscript fully available?

Reviewer #1: No

Reviewer #2: Yes

5. Is the manuscript presented in an intelligible fashion and written in standard English?

Reviewer #1: No

Reviewer #2: Yes

Reviewer #1: The manuscript "Assessment of the influential mechanisms of adolescent risk behaviors and protective and risk factors among high school students in Finnmark, Arctic Norway" has been improved a lot since my late previous review. The theoretical background has been added and it works well. Still, part of my comments or suggestions have not been considered, most importantly the definition of "latent variables". I still feel that this must be taken into account to avoid confusion as here the term "latent variable" has not been used in a standard way as in latent class analysis. In this round I also go more into details.

In its present form the structure of the manuscript is somewhat disordered. Although the journal doesn't have any explicit requirements for section organization, I suggest ordering the manuscript as Follows:

Introduction

Introduction

Background

Alcohol use among Norwegian adolescents (instead of "Alcohol use in Norway")

Antisocial behavior among Norwegian adolescents (instead of "Antisocial behaviour in Norway)

Risk and protective factors for alcohol use and antisocial behavior (instead of "Summary of studies in Norway")

Finnmark’s Unique Context and Adolescent Risk Behaviors

Theoretical and Contextual Framework

Purpose of the Study

- I think the section "Summary of studies in Norway" can be removed as it mostly repeat what is already said in the two previous sections.

Data and methods (instead of Methods)

Data (this includes "Data Source and Samples", "Sample Size" and "Measures")

Statistical methods (this includes "Partial Least Squares Structural Equation Modeling", "Model Development", "Structure

of the Model" and "Data analysis"

Ethics

Results

Evaluation of Measurement Models

Alcohol use

Antisocial behavior

(these two instead of "Evaluation of Structural Models")

Discussion

Environmental Factors

Individual Factors

Adverse Experiences

Leisure time activities (instead of Digital Media Use and Physical Activities as Free Time Use)

Theoretical Implications

Political Implications

Strengths and Limitations (probably there are also strengths in this study?)

Conclusion

- I suggest removing the section "Attitudes and Behaviors" as they are not really measured here.

Specific comments on the Introduction:

p3, l12-15: "Understanding the interplay between individual factors, environmental influences (family, school, peers), and regional contexts—as seen in Finnmark, Norway—can provide insights into how digital and social media shape adolescent health outcomes." - This sentence feels a bit loose as it is not the focus of this study and no answers to this are offered. I suggest removing it.

p8, l10-15: The purpose of the study should be rewritten avoiding terms like "underlying mechanisms" and "influence". As this study uses cross-sectional data, causality can't be studied. So, I suggest using terms like associations or relationships or otherwise avoid referring to causality.

p9, l13-14: The grades may not open up to other than Norwegian readers, present also ages of students.

Specific comments on Data

The description of all variables should be presented under the "Measures" section. Now they are under "Model development" and "Structure of the Model". In section "Data source and samples", also class and gender are mentioned. As all readers may not be familiar with the Norwegian school system, I suggest presenting age together with class.

- The construction of latent variables is not clear. It looks like the individual variables that have been included in the latent variables have been selected by the author and not by using for example latent class analysis (LCA) as suggested by Lanza (e.g. Lanza, S. T., & Rhoades, B. L. (2013). Latent class analysis: an alternative perspective on subgroup analysis in prevention and treatment. Prevention science: the official journal of the Society for Prevention Research, 14(2), 157–168.

https://doi.org/10.1007/s11121-011-0201-1). I suggest finding some other term than "latent" for variables that have been constructed from several variables. In addition part of the "latent" variables are same as individual variables (e.g. SES, religion) and thus can't be considered as latent.

It is also unclear, how the "latent" variables were constructed in practice. For example, on p14 (l10-14) it is stated that "BALKO refers to risky behaviors related to alcohol use and comprises five measured variables, including having consumed any type of alcoholic beverage, having consumed enough alcohol to feel clearly intoxicated, and having driven after drinking alcohol. These behaviors are measured by the variables “balko1” to “balko5." - How is this "latent" variable then created? A sum of all the listed variables? And how it is decided that these variables together form a "latent" variable? The same goes for all the "latent" variables. Please open this up!

- The description of individual variables is not sufficient either, as the response options are not offered. I understand that it is not possible to write out all the used 65 variables (not 92 anymore, p11, l3), but could you offer a reference to the Ungdata questionnaire or if this is not possible, describe the variables (original questions with response options) as a supplement? The description of how socioeconomic status was measured is missing altogether. Finally, it is not necessary to present the variable names (individual and latent) in Table 1.

Finally, I wonder if age was standardized in the analysis? If not, it should at least be discussed as the age range seems to be at least three years (the data includes three class levels). In adolescence, three years (or more?) makes a big difference with regard to the studied outcomes.

The following comments are related to individual variables or their names.

1) "Depression": I would call this construct "depressive symptoms" rather than "depression", which refers to a diagnosis (that is not measured here).

2) "Digital use": In table 1 this is described as "Playing computer games/video games". If this is what the really measures, a better name for this variable would be "gaming". However, on p27, l8-9 digital media use is defined as "online gaming and video watching". So, which one is correct? If also video watching is included, I think it is somewhat questionable to put these two into one question or variable as they are qualitatively quite different. Videos are also watched via social media platforms so this can partly be confused with social media use.

3) "Trust between parents": The name of this variable sounds like it measures trust between parents and not how parents trust their children. In research literature, this is often called e.g. parental monitoring and I suggest using this term also here.

4) "School": I think a more correct name for this would be e.g. "Liking school".

5) "Attitude toward drinking": I think this is particularly problematic, as it really doesn't measure attitude toward drinking but toward drunk driving. In going through the results and discussion, this is however considered as presenting attitude toward alcohol use. This should be corrected throughout the manuscript and also mentioned as a limitation when discussing attitudes (p28, l7-13).

6) "Smoking": as this includes also snus and e-cigarette, I suggest naming this as "Tobacco and nicotine product use" as is the common habit in the research literature.

7) "Drinking": I don't think being a passenger of a drunk driver should be included here as it doesn't measure own alcohol use. I would also consider leaving drunk driving out of this "latent" variable. At least you should test whether the results of the main analysis on alcohol use remain the same if these two are excluded, and if you decide to include them, present some reasoning for this decision. To continue, I'd call this measure "alcohol use" rather than "drinking".

8) "Drug use": As with alcohol use, I don't think being offered cannabis measures drug use. Also here you should test how removing this item affects the results and present reasoning for including this variable if you end up in doing so.

9) "Aggressive behavior": Why is selling drugs included here? Again, test the "latent" variable without this item and explain the inclusion criteria.

Specific comments on Results

- In Tables 2, 3a and 3b, present what is measured instead of variable names. To help reading the tables, the same subheadings could be provided in tables as in Discussion (Environmental Factors, Individual Factors, Adverse Experiences,

Leisure time activities) and the variables grouped under them. The same order could be used in writing out the results.

- Partly related to the description of variables - the presentation of results need clarification. The direction of the associations is mostly not clear as you use expressions like "The final model showed positive associations with religion (RELIG; β = 0.114), depression (DEPR; β = 0.053), COVID-19 experiences (COVID; β = 0.072), sexual harassment (SEXTRAK; β = 0.107), social media use (SMS; β = 0.059), spending evenings with friends (BVENN; β = 0.072), attitude toward drinking (AALKO; β = 0.059), knowing drunk drivers (KALKO; β = 0.210), smoking (BROEYK; β = 0.268), drug use (BDRUG; β =12 0.156), and antisocial behaviors (BANTI; β = 0.155)." (p19, l6-14). For example, did religiousness (not religion) increase or decrease the risk of alcohol use? Please go through all the results paying attention to this.

- I would avoid using variable names in reporting results, but instead use expressions related to what was measured. So, this sentence could be easier to read if reformulated as follows: "A high level of religiousness increased the risk of alcohol use (β = 0.114). Also depression (or depressive symptoms?) increased the risk (β = 0.053)..."

Specific comments on Discussion

p22, l12-13: "Interestingly, peer relationships showed a negative association, suggesting that greater trust in friends correlates with increased alcohol use." - Only one question in the VENN variable is about trust, it seems that this construct rather measures having friends and not trust. Please correct this.

p22, l21: "...descriptive norms of drinking (peers' actual drinking behaviors) have consistently been found to predict adolescent alcohol use. Moreover, descriptive norms significantly mediated the relationship between drinking with peers and alcohol use for both males and females, with a somewhat larger effect observed among males."

- This is somewhat complicated language, maybe just say that peers' drinking behavior is associated with own alcohol use?

p23, l1-2: "injunctive norms (perceived approval of drinking) shaped both immediate and future drinking behavior, highlighting the role of social conformity."

- Again really complicated language, simply perceived (peers?) approval of drinking?

p23, l10-11: "These regional differences highlight the varying impact of family and school environments on adolescent behavior across Norway."

- Where the used measures similar? It appears that no. I would be careful in making conclusions like this with different data and measures.

p24, l1-4: "This dynamic may help explain why spending evenings with friends is strongly associated with multiple risk behaviors in Finnmark. Increased unsupervised time during the evenings may heighten exposure to delinquent peers, thereby increasing the likelihood of engaging in risky behaviors [44], including alcohol use and antisocial activities".

- Maybe because of less possibilities for organized activities than in e.g. the metropolitan area?

p24 (Individual factors): In this chapter there is again a lot of unclear expressions related to the direction of associations. For example "...future outlook showed a negative association with alcohol use". Future outlook -> positive future outlook. This is actually said in other words on p25, l12-13 (repetition), but the expression is more clear here.

p26, l7-8: "...bullying others was associated only with antisocial behaviors".

- Isn't bullying others a type of anti-social behavior? In addition, I don't think it should be considered an "adverse experience" as other experiences under this title refer to something that one has been exposed to and not something that one has been doing him-/herself. Please consider, whether this need to be studied as a separate variable or could it be combined under the outcome "antisocial behaviors.

p27, l1-2: "Finnmark’s isolation and limited activities [48] may explain why alcohol remains a coping mechanism."

- This is quite speculative. This study doesn't offer any evidence of alcohol as a coping mechanism.

p27 (Leisure activities): Again, rewrite this section so that the direction of associations becomes clear.

p27, l12-13: "Since spending time with friends in the evening was associated with increased alcohol use and antisocial behaviors [7, 18], reduced in-person socializing due to digital engagement may explain the inverse relationship."

- But are these negatively associated? So that those who spend a lot of time with peers do not spend that much time on digital activities? Did you test this? What if they watch videos together with friends? I think this reasoning is quite weak and should be rewritten.

p27, l18-19. "36, 37]. "Since social media amplifies peer influences, adolescents may adopt drinking behaviors as a form of social integration."

- Please provide scientific evidence with references for this line of reasoning. It is also argued that time spent in social media is away from time spent with friends irl. As drinking among adolescents mostly is a social activity, more time spent in social media instead of irl would decrease alcohol use.

p27, l24-25: "Regional differences, such as those in Finnmark, where social opportunities are more limited [48], may also contribute to these findings."

- What differences? Maybe regional characteristics? And how would they contribute to your findings. Please open up.

p28 (Attitudes and behavior) Attitudes toward drunk driving is not the same thing as attitudes toward drinking and referring to studies on perceived access of alcohol are really out of context here. The lack of measuring attitudes should rather be discussed in limitations.

p29, l16-18: "The study also supports the theory of planned behavior [59] by showing that attitudes toward alcohol use predict drinking behaviors..."

- As you didn't actually measure this attitudes toward alcohol use but toward drunk driving, I don't think you can say this. In addition, no causal inferences can be made with this cross-sectional data (attitudes predict drinking).

p30, l8: "Recognizing the association between digital media and alcohol use..."

- Social media, not digital media? However, as there is evidence of both negative and positive associations between social media use and alcohol use, this section should be elaborated.

p31, l8: "Attitudes toward alcohol use were identified as a key factor in shaping drinking behaviors, consistent with the theory of planned behavior."

- Again, you didn't measure attitudes toward alcohol use but drunk driving! And I don't see how this is consistent with the theory of planned behavior, as mentioned earlier, causality can't be inferred here. Please remove this conclusion.

p31, l12-13: "Cultural factors, such as Sámi parenting practices emphasizing autonomy, may also contribute to the unique behavioral patterns observed in the region."

- I would be careful in these kinds of interpretations. Looking at Ungdata results, parenting related variables don't seem to be very different in Finnmark compared to other parts of Norway, rather there seems to be variation between communities all around the country, also in Finnmark.

Reviewer #2: The authors responded to all questions/suggestions. Therefore, it is considered that the manuscript is ready to be accepted and published.

**Do you want your identity to be public for this peer review?** For information about this choice, including consent withdrawal, please see our Privacy Policy

Reviewer #1: No

Reviewer #2: **Yes: ** Ana Rita Conde

---

## [Author Response · Author response to Decision Letter 2]

12 May 2025

Response to the Reviewers’ Comments and Concerns

Reviewer #1: The manuscript "Assessment of the influential mechanisms of adolescent risk behaviors and protective and risk factors among high school students in Finnmark, Arctic Norway" has been improved a lot since my late previous review. The theoretical background has been added and it works well. Still, part of my comments or suggestions have not been considered, most importantly the definition of "latent variables". I still feel that this must be taken into account to avoid confusion as here the term "latent variable" has not been used in a standard way as in latent class analysis. In this round I also go more into details.

In its present form the structure of the manuscript is somewhat disordered. Although the journal doesn't have any explicit requirements for section organization, I suggest ordering the manuscript as Follows:

Introduction

Introduction

Background

Alcohol use among Norwegian adolescents (instead of "Alcohol use in Norway")

Antisocial behavior among Norwegian adolescents (instead of "Antisocial behaviour in Norway)

Risk and protective factors for alcohol use and antisocial behavior (instead of "Summary of studies in Norway")

Finnmark’s Unique Context and Adolescent Risk Behaviors

Theoretical and Contextual Framework

Purpose of the Study

- I think the section "Summary of studies in Norway" can be removed as it mostly repeat what is already said in the two previous sections.

Data and methods (instead of Methods)

Data (this includes "Data Source and Samples", "Sample Size" and "Measures")

Statistical methods (this includes "Partial Least Squares Structural Equation Modeling", "Model Development", "Structure

of the Model" and "Data analysis"

Ethics

Results

Evaluation of Measurement Models

Alcohol use

Antisocial behavior

(these two instead of "Evaluation of Structural Models")

Discussion

Environmental Factors

Individual Factors

Adverse Experiences

Leisure time activities (instead of Digital Media Use and Physical Activities as Free Time Use)

Theoretical Implications

Political Implications

Strengths and Limitations (probably there are also strengths in this study?)

Conclusion

- I suggest removing the section "Attitudes and Behaviors" as they are not really measured here.

Response:

Thank you for your helpful suggestions regarding the manuscript structure. In response, I have reordered and renamed the relevant subsection titles according to your proposed outline. I have also removed the sections “Summary of Studies in Norway” as it was repetitive or not directly relevant to the core focus of the study. The section “Attitudes and Behaviors” was also removed—not only based on your suggestions, but also due to the exclusion of variables related to attitudes toward alcohol use (see the detailed response to comments concerning individual variables and their naming).

Specific comments on the Introduction:

p3, l12-15: "Understanding the interplay between individual factors, environmental influences (family, school, peers), and regional contexts—as seen in Finnmark, Norway—can provide insights into how digital and social media shape adolescent health outcomes." - This sentence feels a bit loose as it is not the focus of this study and no answers to this are offered. I suggest removing it.

p8, l10-15: The purpose of the study should be rewritten avoiding terms like "underlying mechanisms" and "influence". As this study uses cross-sectional data, causality can't be studied. So, I suggest using terms like associations or relationships or otherwise avoid referring to causality.

p9, l13-14: The grades may not open up to other than Norwegian readers, present also ages of students.

Response:

Thank you for your valuable comments on the Introduction.

p. 3, l. 12–15 (Version 1): I have removed the sentence as suggested. It was located just before the subheading “Alcohol Use Among Norwegian Adolescents” (p. 3, l. 39).

p. 8, l. 10–15 (Version 1): I have revised the purpose statement to avoid terms such as “underlying mechanisms” and “influence.” Instead, I have used terms like “associations” and “relationships” (see Purpose of the Study, p. 7, l. 22–p. 8, l. 3).

p. 9, l. 13–14 (Version 1): While the Ungdata survey did not include students’ exact ages, I have now indicated the typical age range corresponding to each school grade to clarify this information for an international audience (Data Source and Samples, p. 9, l. 8–12).

Specific comments on Data

The description of all variables should be presented under the "Measures" section. Now they are under "Model development" and "Structure of the Model". In section "Data source and samples", also class and gender are mentioned. As all readers may not be familiar with the Norwegian school system, I suggest presenting age together with class.

Response

Thank you for the insightful and constructive feedback.

Placement of Variable Descriptions (Measures section)

As responded in the previous comments, I have reordered and renamed the relevant subsection titles including “Measures” section (s.9, l.25 – s.14, l.1).

I agree that all variable descriptions should be consolidated under a dedicated "Measures" section for clarity and consistency. In the revised manuscript, the full description of measured and composite variables has been relocated to a newly organized “Measures” section, which now includes all relevant details previously presented across multiple sections (“Model Development” and “Structure of the Model”).

Clarification of School Grade and Age

As responded in the previous section, while the Ungdata survey did not include students’ exact ages, I have now indicated the typical age range that corresponds to each school grade to make the information clearer to an international audience (Data Source and Samples, p. 9, l. 8–12).

- The construction of latent variables is not clear. It looks like the individual variables that have been included in the latent variables have been selected by the author and not by using for example latent class analysis (LCA) as suggested by Lanza (e.g. Lanza, S. T., & Rhoades, B. L. (2013). Latent class analysis: an alternative perspective on subgroup analysis in prevention and treatment. Prevention science: the official journal of the Society for Prevention Research, 14(2), 157–168.

https://doi.org/10.1007/s11121-011-0201-1). I suggest finding some other term than "latent" for variables that have been constructed from several variables. In addition part of the "latent" variables are same as individual variables (e.g. SES, religion) and thus can't be considered as latent.

It is also unclear, how the "latent" variables were constructed in practice. For example, on p14 (l10-14) it is stated that "BALKO refers to risky behaviors related to alcohol use and comprises five measured variables, including having consumed any type of alcoholic beverage, having consumed enough alcohol to feel clearly intoxicated, and having driven after drinking alcohol. These behaviors are measured by the variables “balko1” to “balko5." - How is this "latent" variable then created? A sum of all the listed variables? And how it is decided that these variables together form a "latent" variable? The same goes for all the "latent" variables. Please open this up!

Response

I appreciate the suggestion regarding terminology.

Use of the Term “Latent Variable”

To avoid confusion with statistical latent variable models (e.g., those derived from latent class analysis), we have replaced the term latent variable with composite variable throughout the manuscript. This term better reflects the construction process used in our study, which was theory-driven and based on the thematic grouping of items from the Ungdata survey, not data-driven statistical clustering methods like LCA.

Construction of Composite Variables

The construction of each composite variable has been explicitly described in the revised "Measures" section. For example, the composite variable “alcohol use” consists of three observed indicators. Each item was standardized, and the composite score was computed using the PLS-SEM algorithm, which calculates weighted linear combinations of indicators based on their outer loadings. The rationale for grouping these items was informed by both theoretical constructs and prior empirical applications within the Ungdata framework. These construction steps have been detailed (“Measures” p. 10, l.12-18).

- The description of individual variables is not sufficient either, as the response options are not offered. I understand that it is not possible to write out all the used 65 variables (not 92 anymore, p11, l3), but could you offer a reference to the Ungdata questionnaire or if this is not possible, describe the variables (original questions with response options) as a supplement? The description of how socioeconomic status was measured is missing altogether. Finally, it is not necessary to present the variable names (individual and latent) in Table 1.

Response

Measurement of Socioeconomic Status (SES)

I included a detailed description of how socioeconomic status (SES) was measured. In this study, SES was assessed using an index developed specifically for the Ungdata survey that comprises three key dimensions: parental education, the number of books in the home, and indicators of family affluence (“Measures” p.10, l.19-p.11, l.2).

Description of Individual Variables and Response Options

Given the extensive number of variables used in the analysis, we recognize the impracticality of listing all question wordings and response categories in the main text. In response, Table 1 has been revised to include the original item wordings and corresponding response options (translated into English) for all measured and composite variables used in the model. This approach replaces the earlier version of Table 1, avoiding making a separate supplementary table while ensuring transparency and clarity. Although the complete Ungdata questionnaire cannot be reproduced due to data access restrictions, all items relevant to this study are now presented in Table 1 with appropriate attribution (p.12, l.12 – p.14).

Finally, I wonder if age was standardized in the analysis? If not, it should at least be discussed as the age range seems to be at least three years (the data includes three class levels). In adolescence, three years (or more?) makes a big difference with regard to the studied outcomes.

Response

I acknowledge the concern about the potential confounding influence of age, particularly given the developmental variability across adolescence. Although age was not standardized in the PLS-SEM model due to the absence of direct age data in the Ungdata survey, we have now addressed this limitation explicitly in the “Limitation (p.32, l.23 – p.33, l.4)”. As a proxy, school grade was used, but I recognize that intra-grade age variation may exist and could influence certain outcomes. This limitation has been noted, and recommendations for future data collection practices are offered.

Specific Comments on Variable Naming and Content

The following comments are related to individual variables or their names.

1) "Depression": I would call this construct "depressive symptoms" rather than "depression", which refers to a diagnosis (that is not measured here).

2) "Digital use": In table 1 this is described as "Playing computer games/video games". If this is what the really measures, a better name for this variable would be "gaming". However, on p27, l8-9 digital media use is defined as "online gaming and video watching". So, which one is correct? If also video watching is included, I think it is somewhat questionable to put these two into one question or variable as they are qualitatively quite different. Videos are also watched via social media platforms so this can partly be confused with social media use.

3) "Trust between parents": The name of this variable sounds like it measures trust between parents and not how parents trust their children. In research literature, this is often called e.g. parental monitoring and I suggest using this term also here.

4) "School": I think a more correct name for this would be e.g. "Liking school".

5) "Attitude toward drinking": I think this is particularly problematic, as it really doesn't measure attitude toward drinking but toward drunk driving. In going through the results and discussion, this is however considered as presenting attitude toward alcohol use. This should be corrected throughout the manuscript and also mentioned as a limitation when discussing attitudes (p28, l7-13).

6) "Smoking": as this includes also snus and e-cigarette, I suggest naming this as "Tobacco and nicotine product use" as is the common habit in the research literature.

7) "Drinking": I don't think being a passenger of a drunk driver should be included here as it doesn't measure own alcohol use. I would also consider leaving drunk driving out of this "latent" variable. At least you should test whether the results of the main analysis on alcohol use remain the same if these two are excluded, and if you decide to include them, present some reasoning for this decision. To continue, I'd call this measure "alcohol use" rather than "drinking".

8) "Drug use": As with alcohol use, I don't think being offered cannabis measures drug use. Also here you should test how removing this item affects the results and present reasoning for including this variable if you end up in doing so.

9) "Aggressive behavior": Why is selling drugs included here? Again, test the "latent" variable without this item and explain the inclusion criteria.

Response

We sincerely appreciate your careful review and thoughtful suggestions regarding the naming and content validity of individual variables. Below are our point-by-point responses and corresponding revisions:

1. "Depression"

The term “depression” has been revised throughout the manuscript to “depressive symptoms”, as we agree this more accurately reflects the nature of the construct measured, which does not represent a clinical diagnosis.

2. "Digital use"

We acknowledge that our earlier description was imprecise. The Ungdata questionnaire includes two distinct items: (1) “How much time do you spend playing computer games/video games?”—which we now clearly refer to as gaming, and (2) “How much time do you spend on social media (Facebook, Instagram, etc.)?”—which we refer to as social media use. The previously mentioned inclusion of “video watching” was incorrect and has been removed to avoid confusion, as this activity is not separately measured in the questionnaire and could overlap with social media use. I have revised the terminology throughout the manuscript to ensure consistent and accurate use of these labels and have addressed this limitation in the discussion section.

3. "Trust between parents"

The label “trust between parents” has been revised to “parental monitoring” throughout the manuscript, which more accurately reflects the construct, referring to parental awareness and supervision of their child’s activities, consistent with the literature.

4. "School"

This variable was labeled “liking school” throughout the manuscript, which more precisely captures the intent of the items used and aligns with previous studies on school engagement and well-being.

5. "Smoking"

This variable has been updated to “tobacco and nicotine product use” throughout the manuscript, which encompasses traditional smoking, snus, and e-cigarettes, aligning with contemporary terminology in public health research.

6. “Attitude Toward Drinking (or Attitudes Toward Drunk Driving),” “Drinking (or Alcohol Use),” and “Aggressive Behavior”

6.1. Rewording of Constructs

The construct previously labeled “Drinking” has been renamed “Alcohol Use” throughout the manuscript to more accurately reflect the behavior measured.

“Attitude Toward Drinking” was ultimately removed due to both a conceptual mismatch and the lack of a significant association with alcohol use in the revised model (see Section 6.4 below for further details).

6.2. Exclusion of Specific Items and Reanalysis

In response to the reviewer’s comments, the following items were excluded from thei

---

## [Editor Report · Decision Letter 2]

Assessment of the influential mechanisms of adolescent risk behaviors and protective and risk factors among high school students in Finnmark, Arctic Norway

PONE-D-24-59034R2

Dear Dr. Hansen,

We’re pleased to inform you that your manuscript has been judged scientifically suitable for publication and will be formally accepted for publication once it meets all outstanding technical requirements.

Kind regards,

Jiankun Gong

Academic Editor

PLOS ONE
---

## [Editor Report · Acceptance letter]

PONE-D-24-59034R2

PLOS ONE

Dear Dr. Hansen,

I'm pleased to inform you that your manuscript has been deemed suitable for publication in PLOS ONE. Congratulations! Your manuscript is now being handed over to our production team.

Kind regards,

on behalf of

Dr. Jiankun Gong

Academic Editor

PLOS ONE